# Elemental Analysis and Phenolic Profiles of Selected Italian Wines

**DOI:** 10.3390/foods10010158

**Published:** 2021-01-13

**Authors:** Paola Fermo, Valeria Comite, Milica Sredojević, Ivanka Ćirić, Uroš Gašić, Jelena Mutić, Rada Baošić, Živoslav Tešić

**Affiliations:** 1Dipartimento di Chimica, Università degli Studi di Milano, 20133 Milan, Italy; valeria.comite@unimi.it; 2Innovation Center of the Faculty of Chemistry, University of Belgrade, P.O. Box 51, 11158 Belgrade, Serbia; pantelicm@chem.bg.ac.rs (M.S.); ivankai@chem.bg.ac.rs (I.Ć.); 3Institute for Biological Research “Siniša Stanković”—National Institute of Republic of Serbia, University of Belgrade, Bulevar Despota Stefana 142, 11060 Belgrade, Serbia; uros.gasic@ibiss.bg.ac.rs; 4Faculty of Chemistry, University of Belgrade, P.O. Box 51, 11158 Belgrade, Serbia; jmutic@chem.bg.ac.rs (J.M.); rbaosic@chem.bg.ac.rs (R.B.); ztestic@chem.bg.ac.rs (Ž.T.)

**Keywords:** Italian wines, phenolic compounds, anthocyanins, minerals, metals

## Abstract

The study of the chemical composition of wines is nowadays a topic of great interest because of the importance of this market, especially in Italy, and also considering the numerous cases of falsification of famous and very expensive wines. The present paper focused on the analysis of metals and polyphenols in Italian wines belonging to different provenance and types. At this purpose 20 elements were quantified by inductively coupled plasma optical emission spectrometry (ICP-OES) and ICP mass spectrometry (ICP-MS). Regarding polyphenols, a total of 32 were quantified, among 6 were anthocyanins. Furthermore, in 4 samples (1 rosè and 3 red wines) 42 anthocyanins and related compounds were identified by ultra-high performance liquid chromatography (UHPLC)-Orbitrap MS technique (among these, 6 were also quantified). Non-anthocyanins were determined using UHPLC coupled with a diode array detector and triple-quadrupole mass spectrometer (UHPLC–DAD-QqQ-MS). Total phenolic content (TPC) and radical scavenging activity (RSA) were measured using spectrophotometric methods. The results obtained by elemental techniques were submitted to principal components analysis (PCA) allowing to get information on both geographical and botanical origin of the examined wine samples. Some polyphenols have been detected in higher concentrations only in a certain type of wine, as for example in the case of Grechetto wine. Most of the identified anthocyanin derivatives (pyranoanthocyanins) are formed during the aging of wine by reaction with the other wine components.

## 1. Introduction

The biggest share (43.1%) of grape production belongs to Europe (FAOSTAT, Food and Agriculture Organization of the United Nations) [1]. According to data available on FAOSTAT, for the last twenty years (1998–2018) average annual production of 8.06 millions of t puts Italy in the first place among the top ten world grape producers (FAOSTAT). Among the most important wine-growing regions in Italy there are Veneto, Apulia, Emilia-Romagna and Sicily.

Wine is a rich source of numerous compounds of diverse structure, including polyphenols and minerals. Chemical characterization of wine represents a necessary step in wine authenticity examinations [2]. Since ancient times, wine was proclaimed to be a medicine, while in more recent times scientists confirm that moderate consumption of wine could have beneficial effects on health [3]. From all the grape and wine constituents, perhaps the most studied is the effect of polyphenols on human health. Snopek et al. [3] reported that phenolic compounds present in grape and grape products, including wine, contribute to human health protection via antioxidant, anti-inflammatory, anticancer, antimicrobial, antiviral, cardioprotective, neuroprotective, and hepatoprotective activities. Numerous studies have shown that moderate consumption of wine regardless of alcohol intake can have a positive effect on human health [4,5]. In addition, Kiviniemi et al. [6] showed that moderate amounts of red wine increased coronary flow reserve, which is not the case with de-alcoholized red wines. Synergistic effects of individual polyphenols present especially in red wine, could result in positive impact on human wellbeing, [7]. In addition to the positive effects on human health, the content of phenolic compounds is one of the most important factors in the quality of grapes and wine and have a key role on the oenological quality of wine. In fact, polyphenols are important constituents of wine, as the wine sensory properties, such as color, astringency and bitterness, are directly influenced by the polyphenol composition [8,9]. For example, flavonols are responsible for the color of white wines while anthocyanins give red wine its color [10,11].

Also, as reported by Bora et al. [2], the elemental composition directly affects the qualitative characteristics of the wine (alcohol, total acidity, residual sugar and dry extract).

As regards the various chemical constituents present in wine certainly metals and polyphenols are among those most investigated. Phenolic and elemental compositions largely depend on grape variety, although external factors have a significant influence. Some of these factors are vineyard location (altitude, soil type, sunlight exposure, and geological features), climatic conditions, and degree of grape maturity, viticultural practice, and winemaking conditions [8]. Owing to the above, wines made from the same grape varieties can exhibit differences in the chemical composition and sensory attributes as affected by the abovementioned factors.

Metals affect the organoleptic characteristics of wine, too. Most metals are important for efficient alcoholic fermentation. For example, Ca, K, Mg and Na contribute to regulate the cellular metabolism of yeasts [12]. The metals content in wine has a double origin: natural (or geogenic) origin or due to contamination and/or pollution [12]. Geogenic metals, which can be considered as primary metals, are characteristic and come from the soil on which the vines are grown. From the grape, metals then can reach the wine. This process is linked to some factors such as the ripening of the grapes, the variety and the climatic conditions during the grape growth. These metals represent most of the total metal content in wine. On the contrary, metals of a secondary origin are introduced through external contamination or environmental pollution that reaches wine during grape growth or during the different steps of winemaking (harvesting, bottling and cellaring). The content of metals has been linked both to the study of possible anthropogenic contamination [12,13,14] or to the possibility of obtaining information on geographical origin [15,16], also allowing to identify fraudulent use of DOC wine labels [17].

The aim of the present study was to investigate phenolic profile and elemental composition of a range of wines originated from three Italian wine regions (Veneto, Friuli and Umbria). The selected wines are representative of very famous Italian wines: white sparkling such as Prosecco, white such as Greghetto, Ribolla Gialla and Pinot Grigio and red such as Morcinaia, Sangiovese and Merlot. Elemental analysis, phenolic and profile has been examined allowing highlighting some specific features for the different types of wines.

## 2. Materials and Methods

### 2.1. Analyzed Samples

A total of 13 wine samples from Italy were included in the present study (Table 1). The analyzed wines originate from three Italian regions (Veneto, Umbria and Friuli). All samples were provided by the “Società Agricola Vitivinicola Italiana” (S.AGRI.V.IT.), Sagrivit [18] (https://www.sagrivit.it/lazienda/) which is one of the largest national agricultural realities in Italty. The company manages 14 historic farms, from the north to the south of the Italian peninsula, as well as four wine estates specialized in viticulture for a total of about 5000 hectares of land.

### 2.2. Reagents and Standards

Acetonitrile and formic acid (both MS grade), methanol (HPLC grade), Folin-Ciocalteu reagent, sodium carbonate, sodium hydroxide, hydrogen peroxide, and hydrochloric and nitric acid were purchased from Merck (Darmstadt, Germany). 6-hydroxy-2,5,7,8-tetramethylchroman-2-carboxylic acid (Trolox) molecular sieves (3.2 mm pellets, UOP type 3A), and sodium acetate were purchased from Sigma Aldrich (Steinheim, Germany). 2,2-Diphenyl-1-picrylhydrazyl (DPPH˙) was purchased from Fluka AG (Buch, Switzerland). The Strata C18-E (500 mg/6 mL) SPE cartridges used for the extraction and concentration of anthocyanins from wine samples were obtained from Phenomenex (Torrance, CA, USA). Ultra-pure water (Thermofisher TKA MicroPure water purification system, 0.055 µS/cm) was used to prepare the standard solutions and blanks. Syringe filters (13 mm, nylon membrane, 0.45 µm) were purchased from Supelco (Bellefonte, PA, USA).

Polyphenolic standards (flavonoids aglycones and glycosides, phenolic acids and their derivatives) were purchased from Fluka AG (Buch, Switzerland), while anthocyanin standards were obtained from Extrasynthese (Lyon, France).

### 2.3. ICP Analysis of Elements

For determination of the elemental content in wine (see Table 2), samples were diluted 10 times with water containing 2% (*v/v*) nitric acid (Merck, Germany). Standards were prepared with 1% (*v/v*) ethanol and 2% (*v/v*) nitric acid in order to provide the same concentrations of ethanol and nitric acid as the samples [14,15,16,17,18,19].

The elements were determined using inductively coupled plasma with optical emission spectrometry (ICP-OES, iCAP 6500 Duo Thermo Scientific, Llanthony Rd, Gloucester GL2 8DN, Highnam, UK) and inductively coupled plasma quadrupole mass spectrometry (ICP-QMS, Thermo Scientific, Xseries 2, Hemel Hempstead, UK). ICP-OES was applied for K, Na, Mg, Ca, Rb, and Fe examination, while the remaining 14 elements were analyzed using ICP-QMS. For elemental determination, a multi-element plasma standard solution 4, Specpure, containing 1 g/L of 22 elements were used for calibration [19].

### 2.4. UHPLC–MS Analysis

#### 2.4.1. Preparation of Standard Solutions

A 1000 mg/L stock solution of a mixture of all non-anthocyanin standards was prepared using methanol. Dilution of the stock solution with mobile phase yielded the working solution of concentrations 0.025, 0.050, 0.100, 0.250, 0.500, 0.750, and 1.000 mg/L. Calibration curves were obtained by plotting the peak areas of the standards against their concentration obtaining for all compounds a good linearity, with *R*^2^ values exceeding 0.99 (peak areas vs. concentration).

Anthocyanin standards were prepared using the same procedure as for the non-anthocyanins, except working solutions were diluted in acidified methanol (pH = 2, HCl).

#### 2.4.2. UHPLC–DAD-QqQ-MS Analysis of Non-Anthocyanins

Wine samples were filtered and analyzed without dilution. Prior to UHPLC–DAD-QqQ-MS analysis, the extracts were filtered through a 0.45 µm nylon membrane filter.

The separation, determination, and quantification of the non-anthocyanins in the wine samples were performed using a Dionex Ultimate 3000 UHPLC system equipped with a diode array detector (DAD) that was connected to TSQ Quantum Access Max triple-quadrupole mass spectrometer (Thermo Fisher Scientific, Basel, Switzerland). All experimental condition of UHPLC separation is given in Pantelić et al. [20].

A TSQ Quantum Access Max triple-quadrupole mass spectrometer equipped with a heated electrospray ionization (HESI) source was used for detection of compounds of interests. The parameters of ion source and the other MS conditions are given in our previous work [20]. Xcalibur software (version 2.2) was used for instrument control. The non-anthocyanins were quantified by direct comparison with commercial standards. The total amounts of each compound were expressed as mg/L of wine.

#### 2.4.3. UHPLC-LTQ OrbiTrap MS Analysis of Anthocyanins and Anthocyanin-Derived Pigments

All wine samples were filtered through a 0.45 µm nylon membrane filters before solid phase extraction (SPE). First, C18 cartridges were preconditioned with 5 mL of acidified methanol (pH = 2, HCl) following 5 mL of 0.1% aqueous HCl (pH = 2, HCl). Then, 2 mL of wine were applied. Cartridges were washed with 5 mL of 0.1% aqueous HCl (pH = 2) in order to remove retained sugars, acids, and other water-soluble compounds. After drying with nitrogen gas for five minutes, adsorbed anthocyanins were eluted from C18 cartridges with 1 mL of acidified methanol (pH = 2, HCl). Prior to LC/MS analysis extracts were stored in the dark at 4 °C.

In order to identify and quantify anthocyanins on wines, UHPLC system consisting of a quaternary pump 600 Accela and Accela Autosampler connected to a LTQ Orbitrap mass spectrometer with electrospray ionization-heated probe (HESI-II, Thermo Fisher Scientific, Bremen, Germany) was used. Ion source settings, mass analyzer parameters and chromatographic conditions were the same as in Pantelić et al. [20].

Xcalibur software (version 2.1) was used for instrument control and data analysis. Molecule editor program, ChemDraw (version 12.0) was used as to calculate accurate mass of compounds of interest. Unknown compounds were identified on the basis of their monoisotopic mass and MS^4^ fragmentation, and confirmed using previously reported MS fragmentation data. Full scan analysis was employed for detection of the monoisotopic masses of unknown compounds, while the MS^4^ experiment provided fragmentation pathways.

### 2.5. Determination of Total Phenolic Content (TPC)

Total phenolic content in samples was determined using a modified version of the Folin-Ciocalteu method described in literature [20]. Briefly, each wine sample (0.5 mL) was mixed with 0.5 mL of ultrapure water and 2.5 mL of Folin-Ciocalteu reagent (100 mL/L) After 5 min, 2 mL of sodium carbonate (75 g/L) was added and the mixtures were left to incube during 2 h (at room temperature, in the dark). After incubation, the measurements of the absorbance (at 765 nm) were performed on a Cintra 6 UV-VIS spectrometer (GBC Scientific Equipment Ltd.). A mixture of water and reagent was used as a blank, while gallic acid (in the range of 20−100 mg/L) was used as a standard for the calibration curve construction. TPC values were expressed as gram of gallic acid equivalent (GAE) per L of wine (g GAE/L). All measurements were done in triplicate and the results were expressed as mean values.

### 2.6. Determination of Radical-Scavenging Activity (RSA)

Radical scavenging activity was determined using *DPPH* radical solution by a slightly modified literature methods [15]. The extracts (0.1 mL) were mixed with 4 mL of methanol solution of *DPPH*˙ (71 µmol/L), and the mixtures were left to stand for 1 h in the dark, at room temperature. The absorbance was measured at 515 nm against blank cell filled with methanol, on Cintra 6 UV-VIS spectrometer. *RSA* was calculated as a percentage of *DPPH˙* discoloration using the equation:$$RSA\;(\%)\;=\;\frac{\left(A_{DPPH}\;-\;A_{sample}\right)}{A_{DPPH}}\;\times \;100$$
where *A_DPPH_* is the absorbance of pure *DPPH*˙ solution and *A_sample_* is the absorbance of *DPPH˙* solution in the presence of samples. Trolox was used as standard (concentrations ranged from 100 to 600 µmol/L) and calibration curve was displayed as a function of the percentage of *DPPH* radical inhibition (*RSA* (%)). The results were expressed as millimoles of Trolox equivalents (TE) per L of wine (mmol TE/L). The results were presented as mean values of three measurements.

### 2.7. Data Analysis

Data analysis has been carried out by means of Statistica 8.0 (Stat Soft) software package which was used for Principal Component Analysis (PCA), means with error plots, box and whiskers plots and for the calculation of the correlation matrix. Anova one-way and F test were performed by Excel (Microsoft).

## 3. Results and Discussion

The main goals of the present study are to achieve information on the specific characteristic of the analyzed wines trying to highlight the differences through the combined application of analytical methodologies and data treatment by statistical methods.

### 3.1. Major and Trace Elements in Italian Wines

As expected, K was the most abundant element in all wine samples (Table 2), ranged between 488.7 and 1174.0 mg/L. In Appendix A, the data with the corresponding standard deviations are shown. These amounts were in accordance with results available in literature [19,21,22]. The next by the abundance were magnesium and calcium, found in similar quantities from 52.44 to 102.00 mg/L and from 50.73 to 111.20 mg/L, respectively. If compared with other data reported in the literature, these values were in agreement with the results obtained for Spanish [22] and Serbian [19] wines. On the other hand, Ca and Mg contents were higher when compared to data reported for wines from Belgium [23]. Na content ranged from 7.7 mg/L (Asolo Preosecco Brut) to 38.2 mg/L (Belfiore), that was lower if compared to Spanish red wines [22], but in agreement with the results published on Serbian [19] wines.

Concentrations of Rb are usually higher in red wines compared to white wines [19]. This was confirmed in the present study, where average content of Rb in white wines and rosè was 3.63 mg/L, while in red wines it amounted 6.03 mg/L. The values obtained for Rb indicated that the studied Italian wines are richer in rubidium than Belgian [23], Czech [21] and Romania [24] ones. Higher concentrations of Rb in wines could be associated with the composition of the soil on which the vines were cultivated [19]. Moreover, higher contents of Mg and Ba were observed in rosè sparkling wine (W4) and red wines (W7, W8 and W13) in comparison to white wines. The average content of Fe (0.59 mg/L) indicated that the studied Italian wines are moderately rich in Fe, having less Fe than Turkish [25] or Croatian [26] wines, but more than Argentinian [27] ones.

Elements such as Al, Cu, Mn, and Zn were present in amounts similar to previously published results for Croatian [26], Romanian [24] and Turkish [25] wines.

For better visibility standard deviations are given in Appendix A.

The amounts of toxic elements (Pb, As and Cd) were below the permitted values (0.2 mg/kg for Pb and As, and 0.01 mg/kg for Cd) in all tested wines (Commission regulation (ec) no. 466/2001, Off J Eur Communities 2001; L77/9). It should be stated that sample W8 stood out with particularly high content of Pb, 175.04 μg/L, which is close to the maximum allowable value. In the other samples, Pb contents were in the range 0.24–79.97 μg/L, which was significantly lower than maximum allowed. As for arsenic, in the sample W3 its content was 19.84 μg/L, while the concentrations for all the other samples were set equal to LOD value, i.e., 0.2 ppb. The average content of Cd was 0.79 μg/L and in any case was below the permitted limit. Cu was also present in allowed quantities, bellow 1 mg/kg (International Office of Vine and Wine (OIV)). The contents of Cu in investigated wines were in the range from 0.11 to 97.46 μg/L, with the exception of sample W2, where concentration of Cu was notably higher (469.49 μg/L), but still below the limit.

In order to get information on the origin of wine in terms of geographical provenance principal component analysis (PCA) were successfully applied in the literature [15,16] while to highlight how specific condition (weather or harvest year) influence some wine properties such as the taste, other statistical methodologies including ANOVA and the Tuckey test were employed [28].

In order to verify a possible correspondence between the geographical origin, botanical origin, wine type and the chemical composition, the results on chemical composition (Table 2) were submitted to principal component analysis (PCA) considering at first all the elements as input variables. It is worth noting that multivariate methods are commonly applied for wines provenance studies [15,16,17,29,30,31,32].

The scatter plots obtained considering the first two components were reported in Appendix A. No obvious groupings were observed based on the geographical origin neither on the base of the wine type. Instead, some wine samples were grouped based on the concentrations of some elements, as it was evident from the loading plot reported in Appendix A. In particular, samples W8, W9, W10 and W13 were characterized by high Mn values. Wines W4, W7, W8, W10, W12, and W13 were richer in K in comparison to the remaining samples. Moreover, the amounts of Zn were higher in samples W2 and W5 when compared to the other wine samples.

Taking into account the high loading values of Mn and Zn on the first two components (Appendix A), in order to try to disentangle more information, these two elements were excluded from the data set and PCA again carried out. At this point, on the base of the new results obtained from PCA, K and Al were the elements with the highest loadings (see at this purpose the loading plots shown in Appendix A). Therefore, they were excluded and PCA was carried out for the third time on the new data set which at this point was formed by all the analyzed elements excluding Mn, Zn, As, Al and K (As was excluded since all the samples with only one exception had a values set equal to LOD as mentioned before). The results obtained, reported in Figure 1a,b, finally were informative regarding the geographical origin of the wine samples (Figure 1a). In fact, considering the first 3 components, that together account for about 70% of the system variance, a rather good separation among the wines coming from Veneto, Umbria and Friuli was noticed. It is worth noting that the elements responsible for the separation of wines along PC1 were Mg, Ba, Rb, Se and Fe (Appendix A). Those are elements of geogenic origin [12,17] and this indicates that soil composition is a crucial variable to differentiate wines on the base of their geographical origin.

Within the same geographical region, information related to the botanical origin was also achievable (Figure 2b). Indeed, in the Friuli wines group, samples W9 and W10, both Pinot Grigio, were close to each other and more distant from the other Friuli wines; among these, W11 and W12 (Ribolla Gialla) were very close together. It is worth noting that in order to check these observations, *F* test was carried out comparing the average composition obtained for Pinot Grigio wines with the average composition obtained for Ribolla Gialla and they resulted significantly different (*F* = 1.86 > *F*crit = 1.82, *p* = 0.05).

Furthermore, among Umbria wines, samples W5 and W6 (Grechetto) were also closer together with respect the other wines belonging to this group.

A different approach, based on means with error plots (Figure 2, Figure 3 and Figure 4) was also applied to evidence differences among Veneto, Friuli and Umbria regions. As regards minor elements (Figure 2a, Figure 3a and Figure 4a), Umbria wines show the highest concentration of Ba followed by Friuli wines. Barium could be associated to the geochemical features of the soil on which the vineyard was grown (at this purpose see also the loading plot reported in Appendix A where Ba was well correlated with the other elements of geogenic origin). Friuli and Umbria wines were characterized by slightly higher Mn concentrations with respect to Veneto wines, which instead had higher Cu value (Figure 2a). Cu concentration is an efficient variable for differentiating wines according to the applied enological treatment [12].

Umbria wines were also characterized by the highest Pb concentration (Figure 4a). As human intakes Pb mainly through food consumption (vegetables, cereals and beverages, particularly wine), some studies [14,33] regarding Pb exposure through dietary in an Italian community were recently carried out. The average lead concentration reported for wine is of about 14 ppm, with higher values for red wines (up to 44 ppm) [33]. Fortunately, these two studies concluded that the estimated intakes were below the tolerable upper intake levels, so the levels of trace elements in diet of the investigated population could be considered safe. The highest concentration of Pb was determined in red wine W8 (Morcinaia, Umbria), 175.04 µg/L, while in the other wine samples it was present in lower values (0.24–79.97 µg/L).

Regarding the main elements (Figure 2b, Figure 3b and Figure 4b), as already highlighted, K was the most abundant one. In order to relieve comparison among the other elements present in lower concentrations, K was excluded from the graphs. The resulting graphs (Figure 2c, Figure 3c and Figure 4c) revealed that Ca concentrations were similar for the three regions; content of Mg was slightly higher in Umbria wines, while Veneto wines showed lower Na concentrations.

Another method for representing data trends are box and whiskers plots. These kinds of plots are reported for Veneto, Umbria and Friuli wines in Appendix A respectively. It is worth noting that what observed in Figure 2, Figure 3 and Figure 4 is basically confirmed by the box and whiskers plots and no additional information is achieved with the exception that it is more evident that Friuli and Veneto wines show a slightly higher concentrations of Sb and Ni with respect to Umbria wines.

In order to establish possible interactions among the elements, the correlation matrix was generated (Appendix A). The most significant correlations with *p* < 0.005 were observed among Ba, Pb, Se, Fe, Mn, Mg, and Rb (correlation factors (r) were between 0.75 and 0.93). When the level of significance was changed to *p* < 0.05 some additional elements showed statistically significant correlations (K, Co, Cr, and Zn) with *r* ranging from 0.56 to 0.70. It is worth noting that lead showed significant correlations with elements of geochemical origin, such as Se, Rb and Ba [13,17]. In particular, the correlation between Ba and Rb (*r* = 0.88, *p* < 0.005) has been attributed to the same geochemical origin. Elements such as Fe, Cr, Co, Mn had a fairly good correlations and a common anthropogenic origin can be hypothesized. Fiket et al. [13] reported that the presence of some elements in wines (manganese, copper, lead and zinc) could be associated with the application of pesticides, fungicides and fertilizers in vineyards. Furthermore, metals contamination could also come from the winemaking process [12,13].

### 3.2. Phenolic Profile of Italian Wines

Phenolic compounds (non-anthocyanins and anthocyanins) found in Italian wines, RSA and TPC are depicted in Table 3 (in Appendix A the data with the standard deviations obtained are reported). The results obtained herein were compared with previously obtained values for other Italian wines. First of all, it is worth to notice that in the present study, a much larger number of compounds were analyzed including anthocyanins, phenolic compounds for which there is not so many data in the literature.

#### 3.2.1. Spectrophotometric Determination of TPC and RSA

TPC results for sparkling wines were in range 1.15–1.44 g GAE/L while for white wines were between 0.81 and 1.67 g GAE/L. When compared to results for white wines from Montenegro (0.20–0.423 g GAE/L) [34] and Republic of North Macedonia (0.17–0.43 g GAE/L) [35], investigated Italian wines had a higher content of total phenolics. Tourtoglou al. 2014 [36] published mean value of 0.32 g GAE/L for total phenolic contents in Greek white wines, which is lower compared to our results. The highest concentrations of polyphenols (1.71–2.36 g GAE/L) were found in investigated red wines. Similar values were reported for Montenegrin red wines: 1.97–2.67 g GAE/L [7,8] and Italian red wines 1.30–2.76 g GAE/L [37]. Somewhat higher results were published by Raičević et al. [38], Ivanova-Petropulus et al. [39] and Iorizzo et al. [40] for red wines from Montenegro, Republic of North Macedonia and Italy.

As for the radical-scavenging activity of wines (RSA), data published earlier are in accordance with results presented herein. Results for antioxidant activity were in the range 0.39–0,44 mmol TE/L for sparkling white wines, 0.82–3.17 mmol TE/L for white wines and 12.03–18.71 mmol TE/L for red wines. Similar values were reported for red wines from Montenegro [9,41] and Republic of North Macedonia [39]. For Serbian red wines Majkić et al. [42] published lower values in the range 2.30–6.53 mmol TE/L. The same authors for Italian red wine Merlot obtained RSA value of 6.69 mmol TE/L, while for wines originated from France, Spain, Slovenia and Macedonia results were higher than 10.3 mmol TE/L. Mitrevska et al. [35] reported lower RSA values than those presented herein: 4.11–11.73 mmol TE/L for red wines and 0.51–1.30 mmol TE/L for white wines originated from Republic of North Macedonia. Tuberoso et al. [37] reported RSA in the range 8.4–13.2 mmol TE/L for Carignano wines produced in Sardinia, which was lower compared to currently analyzed Italian red wines.

Considering TPC and RSA, it could be observed that these values were higher in red wines than in white wines.

#### 3.2.2. Quantification of Polyphenols

A total of twenty-eight non-anthocyanins and six anthocyanins (i.e., 32 polyphenols) were identified and quantified in examined wine samples (Table 3). Among the identified compounds, gallic, ellagic, gentisic and protocatechuic acids belong to hydroxybenzoic acids; chlorogenic, caffeic, *p*-coumaric and sinapic acid belong to hydroxycinnamic acids; gallocatechin, catechin, gallocatechin gallate and epigallocatechin gallate belong to flavanols; myricetin, rutin, hyperoside, astragalin and galangin belong to flavonols; luteolin, apigenin, cymaroside, apigetrin and chrysin belong to flavones. Phenolic acids were the most abundant phenolic compounds in all investigated wine samples with the exception of sinapic acid, detected only in sample W12. It was reported that sinapic acid has various beneficial effects on human health such as anti-inflammatory, antioxidant, antibacterial and anticancer activities [43]. Protocatechuic, caffeic, *p*-coumaric and ellagic acids were found in all investigated samples, while gallic, chlorogenic and gentisic acids were found in almost all samples. Gallic acid was the most represented phenolic compound in red wine samples (74.36–81.35 mg/L), followed by ellagic acid (24.41–29.02 mg/L). Gallic acid content found in red wines was higher compared to Serbian and Montenegrin red wines [7,42], but lower when compared with Macedonian red wines [44]. It was reported that ellagic acid from red wine in combination with celastrol (a bioactive compound derived from traditional Chinese medicinal herbs) causes in vivo tumor suppression and is recommended to be used as chemoprotective agens for lung cancer [45]. Among hydroxybenzoic acids, gentisic acid was the most abundant in white sparkling wines (concentrations up to 3.36 mg/L), while in rosè sparkling wine (W4) it was gallic acid (41.73 mg/L). Regarding the white wines (non-sparkling), the most abundant hydroxybenzoic acid was gallic acid. For samples, W9–W12 concentration of gallic acid was 2.53–4.40 mg/L, while in samples W5 and W6 it amounted 38.20 and 24.22 mg/L respectively. Somewhat lower gallic acid concentrations were published for Montenegrin (0.9–2.4 mg/L) [34], Greek (0.60–1.69 mg/L) [36] and Italian white wines (0.6–3.5 mg/L) [46]. Samples W5 and W6 were distinct from other white wine samples, by higher content of *p*-coumaric acid, caffeic acid and catechin, as well as gallocatechin presents (only in these two samples). The highest contents of caffeic acid were detected in red wine samples and in white wine samples W5, W6 and W9. The content of *p*-coumaric acid in all wine samples was also considerable (1.12–10.34 mg/L). Regarding flavanols, catechin was the most abundant. In white wine samples W5 and W6 concentrations of catechin were 5.95 and 10.40 mg/L, respectively, while in red wines it ranged from 16.98 to 26.67 mg/L. Similar catechin concentrations were reported for Serbian (9.1–49.3 mg/L), and some other European (11.3–32.2 mg/L) red wines [42], as well as for Montenegrin red wines 9.85–24.35 mg/L [4]. Among flavonols, galangin was found in most of the examined samples in concentration up to 1.21 mg/L. Resveratrol, the only compound from stilbene group, was detected exclusively in red wine sample W7 in concentration 4.00 mg/L. The least represented phenolic compounds in investigated Italian wines was sinapic acid found in W12, luteolin found in sample W8 and chrysin found in W1.

The content of non-anthocyanins is one of the main factors affecting the quality of grapes and corresponding wines. The quantity and structure of non-anthocyanins significantly influence the oenological properties of the grapes such as color, bitterness and stability [8].

In Figure 5 the trends of all polyphenols quantified in the analyzed Italian wines are depicted and, as one could see, distinguishing among sparkling, white and red wines, was achieved.

It is evident how the first 16 polyphenols seem to be the most discriminant among the groups. At this purpose, ANOVA one-way was applied on the data set formed by the 3 categories and on the base of the first 16 polyphenols a significant difference among the 3 categories was obtained (*p* = 0.15).

PCA was then performed on the data reported in Table 3 and the results acheived are shown in Figure 6, putting in evidence how Veneto wines are those more distinguishable from the point of view of the phenolic profile; furthermore, as expected, red wines are well separated.

### 3.3. Identification of Anthocyanins in Red Wine Samples

UHPLC-Orbitrap MS analysis of anthocyanins allowed to quantify 6 derivatives in all the wines considered in this study. Furthermore, in one rosè wine (W4) and three red wines (W7, W8, and W13), in a positive ionization mode, the identification of 42 related compounds was performed (Table 4). Most of the identified compounds belonged to malvidin derivatives (22 compounds), followed by peonidin, deplhinidin, and petinidin derivatives (5 compounds each). Only three cyanidin derivatives were found, and from the group of anthocyanidins (anthocyanin aglycones), only cyanidin and peonidin were identified.

Among all identified anthocyanins, only six (see Table 3) were confirmed using standards, while the other 36 were identified by exact mass search of M^+^ molecular ions and evaluation of its MS spectra (MS^2^, MS^3^, and MS^4^ fragmentation) as well as by comparison with the available literature. The compound numbers and names, molecular formulas, high resolution mass data (calculated and exact masses (M^+^, *m/z*) and mass accuracy errors (ppm)), as well as presence of compound in the samples are summarized in Table 4, while the retention times (*t*_R_, min) and fragmentation data are presented in Appendix A. From Table 4 it can be concluded that the presence of anthocyanins in the rosè wine sample is significantly less than in the red wine samples. Two isomers of petunidin 3-O-glucoside (compound 6 and 34) were found in investigated samples and it is interesting that the first derivative (4.88 min) was found in all four wine samples, while the second (7.10 min) derivative was found only in a sample of rosè wine. In addition to anthocyanidin glycosides, a significant number of acyl-glycosides derivatives with acetyl and comaroyl residue were found. These acyl derivatives are known to be present in red wines [47]. However, the largest number of found derivatives belongs to the group of pyranoanthocyanins, which are formed by the aging of wine. During the aging, the concentration of anthocyanins in wine decreases dramatically due to decomposition, polymerization and reaction with other components of wine, which leads to the formation of compounds derived from anthocyanins (pyranoanthocyanins) [48]. Some of the substances reported, for instance peak 13 have been previously identified [49]. All of these derivatives were previously identified in the wine of grapevine variety Vranac (*Vitis vinifera* L.) from Montenegro [9].

## 4. Conclusions

In the present paper Italian wines coming from three different regions were analyzed with regard to elemental composition and polyphenolic content. Principal component analysis (PCA), applied on elemental composition data, revealed discrimination of 3 wine groups according to the geographical provenance. Geogenic elements, which are characteristic of the soil where the vine was grown, had the greatest contribution to the separation. Within each regional group a differentiation on the base of the botanical origin was also evidenced. With regard to the main elements, K was the most abundant one in all the samples. Some differences were highlighted among the regions on the base of the minor elements: Umbria wines had the highest content of Ba and Pb, while Veneto wines showed the highest Cu concentrations. TPC and RSA were higher in red wines when compared to white wine samples. Among the analyzed wines, Grechetto had a characteristic profile of polyphenols showing higher contents of *p*-coumaric acid, caffeic acid and catechin, as well as gallocatechin, which was present only in this type of wine. Resveratrol was detected only in the red wine from Friuli (Merlot). Sinapic acid, which has various beneficial effects on human health, was revealed only in one sample of Ribolla Gialla while ellagic acid, a chemoprotective agent for lung cancer, was found in significant higher concentration in the red wines. As far as anthocyanins, the quantification of 6 compounds was performed for all the examined wines (their concentration was higher in red wines), while for the rosè and the 3 red wines overall 42 anthocyanins and related compounds were identified, most of them coming from the aging of wine. Overall, the research conducted has made it possible to add new knowledge in the field of the study of the oenological characteristics of Italian wines. Hopefully the number of wines samples could be increased in the future in order to further investigate the aspects related to both origin and enological characteristics.

## Figures and Tables

**Figure 1 foods-10-00158-f001:**
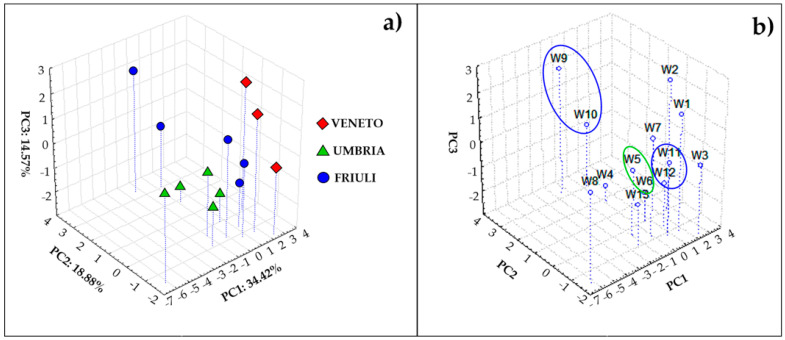
Scatter plot (for PC1, PC2 and PC3) obtained from PCA carried out on the element concentrations excluding from the calculation Mn, Zn, As, Al and K.; (**a**) the 3 Italian regions are indicated; (**b**) the wines names are included and some similarities discussed in the text are evidenced.

**Figure 2 foods-10-00158-f002:**
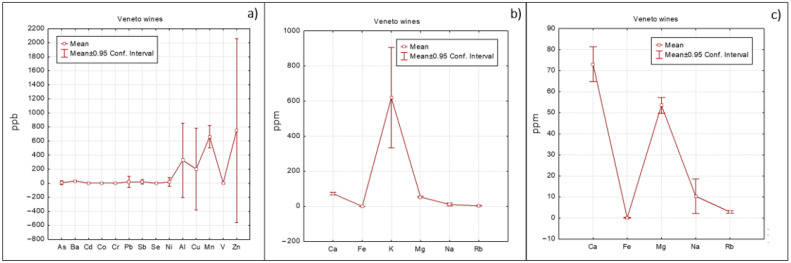
Elements trend concentration for Veneto wines: (**a**) minor elements, (**b**) major elements and (**c**) major elements excluding K.

**Figure 3 foods-10-00158-f003:**
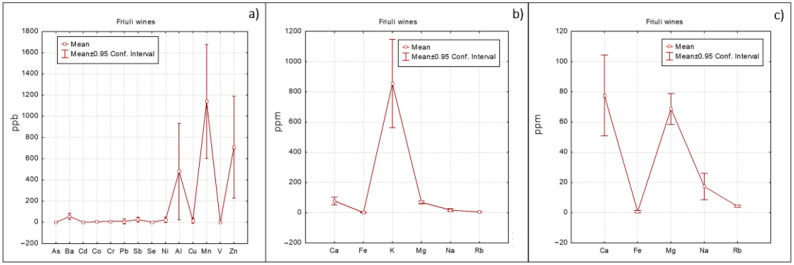
Elements trend concentration for Friuli wines: (**a**) minor elements, (**b**) major elements and (**c**) major elements excluding K.

**Figure 4 foods-10-00158-f004:**
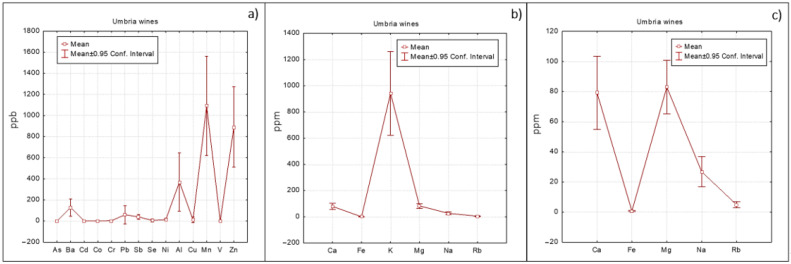
Elements trend concentration for Umbria wines: (**a**) minor elements, (**b**) major elements and (**c**) major elements excluding K.

**Figure 5 foods-10-00158-f005:**
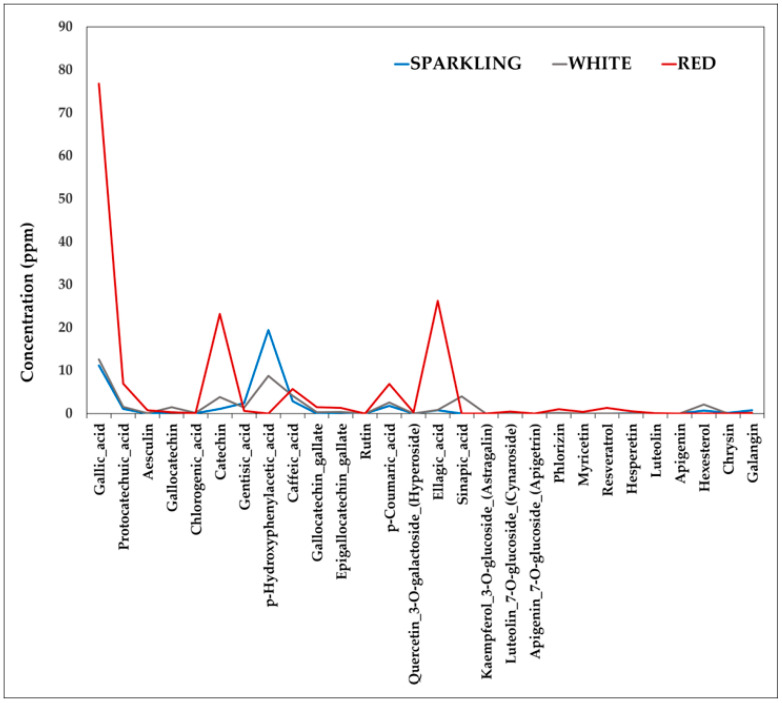
The trends of all the non-anthocyanins determined in the analysed Italian wines distinguishing among sparkling, white and red wines.

**Figure 6 foods-10-00158-f006:**
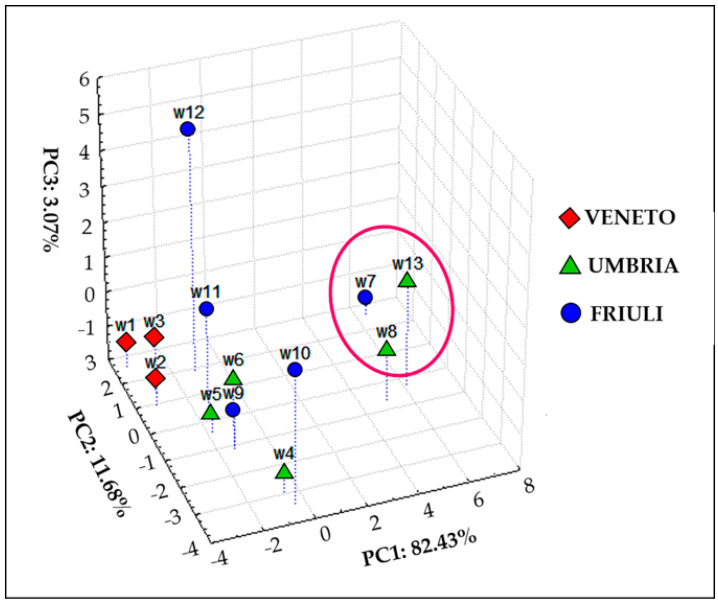
Scatter plot (for PC1, PC2 and PC3) obtained from PCA carried out on the 32 polyphenol concentrations.

**Table 1 foods-10-00158-t001:** Analyzed wine samples; the different characteristics are reported for each wine.

Sample Sign	Wine Sample	Type of Wine	Harvested Year	Botanical Origin	Geographical Region	EtOH Vol (%) ^a^
W1	Asolo Preosecco Brut	white sparkling wine	2014	Prosseco	Veneto	11.5
W2	Asolo Preosecco Extra Dry L6015	white sparkling wine	2014	Prosseco	Veneto	11.5
W3	Asolo Preosecco Extra Dry L2115	white sparkling wine	2014	Prosseco	Veneto	11.5
W4	Belfiore	rosè sparkling wine	2014	Gamay	Umbria	12.0
W5	Grechetto Montenerone	white	2013	Grechetto	Umbria	13.5
W6	Grechetto Montenerone	white	2014	Grechetto	Umbria	13.5
W7	Merlot	red	2013	Merlot	Friuli	14.0
W8	Morcinaia	red	2008	Merlot, Cabernet Sauvignon, Sangiovese	Umbria	14.0
W9	Pinot Grigio	white	2013	Pinot Grigio	Friuli	14.0
W10	Pinot Grigio	white	2014	Pinot Grigio	Friuli	14.0
W11	Ribolla Gialla	white	2013	Ribolla Gialla	Friuli	12.5
W12	Ribolla Gialla	white	2014	Ribolla Gialla	Friuli	12.5
W13	Sangiovese	red	2014	Sangiovese	Umbria	13.0

^a^ The values were stated on the bottle declaration.

**Table 2 foods-10-00158-t002:** Elements quantification in the analyzed Italian wines; the concentrations are reported as µg/L and were determined by ICP-MS with the exception of the elements indicated as ^a^ where the concentrations are reported as mg/L and the analyses were carried out by ICP-OES.

	W1	W2	W3	W4	W5	W6	W7	W8	W9	W10	W11	W12	W13
As	0.20	0.20	19.84	0.20	0.20	0.20	0.20	0.20	0.20	0.20	0.20	0.20	0.20
Ba	32.64	33.87	26.12	123.30	79.80	79.10	83.33	239.70	58.16	77.70	32.07	31.66	122.10
Cd	0.37	1.55	0.94	1.25	0.39	1.37	0.73	0.05	0.98	1.64	0.01	0.59	0.38
Co	3.64	2.09	0.98	4.40	2.77	1.83	2.42	2.68	7.34	7.77	2.19	1.61	2.60
Cr	0.32	0.32	0.32	0.32	4.32	3.03	0.32	6.42	15.41	14.82	11.12	9.16	3.11
Pb	0.24	55.97	0.24	20.79	79.97	11.68	2.24	175.04	45.22	0.24	0.83	0.54	14.90
Sb	32.81	8.59	19.13	18.57	67.54	37.91	9.58	46.50	44.88	18.50	7.26	44.26	29.60
Se	0.14	0.14	0.14	0.14	1.45	0.30	0.23	25.74	2.87	2.06	2.91	2.38	2.99
Ni	4.17	46.01	0.57	30.95	15.34	5.77	35.15	13.86	50.57	17.95	10.00	0.20	5.00
Al	565.70	260.10	155.30	719.80	446.90	290.70	26.55	155.40	443.60	329.00	1027.00	567.70	236.20
Cu	97.46	469.49	36.30	10.07	0.11	0.11	0.11	0.11	21.19	7.73	0.81	49.32	50.21
Mn	735.00	613.49	642.26	1016.26	761.67	742.74	838.77	1633.76	1732.28	1472.22	887.46	771.02	1299.22
V	4.18	1.89	0.57	0.17	1.53	0.17	1.95	3.18	3.24	2.30	0.81	0.73	0.17
Zn	468.40	1359.00	423.80	622.20	1356.00	607.50	298.20	969.60	1278.00	915.60	518.60	536.90	897.50
Ca ^a^	70.73	76.93	71.71	111.20	62.51	83.94	50.73	67.51	62.77	98.10	76.45	99.86	71.11
Fe ^a^	0.13	0.08	0.05	0.79	0.53	0.52	0.59	1.08	1.81	0.69	0.37	0.20	0.87
K ^a^	488.70	706.80	662.70	1152.00	608.70	718.70	965.20	1119.00	567.30	867.40	700.30	1174.00	1111.00
Mg ^a^	55.29	52.87	52.44	77.39	70.30	71.04	82.11	102.00	64.03	68.38	69.10	60.38	94.27
Na ^a^	7.75	14.13	9.32	38.24	29.61	27.41	11.69	16.88	21.60	26.49	9.36	17.62	22.08
Rb ^a^	2.69	2.96	3.08	3.83	3.90	3.85	5.07	7.38	4.33	3.96	4.08	3.59	5.63

**Table 3 foods-10-00158-t003:** Quantitative data on individual phenolic compounds (non-anthocyanins and anthocyanins) in Italian wines, radical-scavenging activity (RSA), and total phenolic content (TPC).

Non-Anthocyanins (mg/L)	W1	W2	W3	W4	W5	W6	W7	W8	W9	W10	W11	W12	W13
Hydroxybenzoic acids													
Gallic acid	1.99	1.09	−	41.73	38.20	24.22	74.36	81.35	4.40	2.53	2.82	3.24	74.62
Protocatechuic acid	0.53	0.57	0.57	2.63	1.33	1.78	7.27	5.41	1.70	1.41	1.57	1.30	8.23
Gentisic acid	1.45	3.24	3.36	1.52	2.90	1.47	1.01	0.87	1.47	0.49	0.71	1.26	−
Ellagic acid	0.71	0.51	0.65	1.17	0.81	1.01	25.24	24.41	1.36	0.74	0.49	0.61	29.02
Hydroxicinnamic acids													
Sinapic acid	−	−	−	−	−	−	−	−	−	−	−	23.67	−
Caffeic acid	0.59	0.67	1.36	8.79	9.16	5.47	4.58	7.18	5.78	2.58	1.01	0.95	5.44
Chlorogenic acid	0.04	0.03	0.04	0.04	0.07	0.10	−	0.07	0.03	0.19	0.12	0.10	0.20
p-Coumaric acid	1.12	2.03	1.46	2.64	3.02	2.99	10.34	4.63	1.80	1.65	1.60	4.30	5.61
Flavanols													
Gallocatechin	−	−	−	1.34	3.29	5.37	−	−	−	−	−	−	0.93
Epigallocatechin gallate	−	−	0.86	−	−	−	1.02	1.63	−	0.91	−	1.48	1.31
Catechin	−	−	−	4.23	5.95	10.40	25.93	16.98	3.63	−	1.49	1.69	26.67
Gallocatechin gallate	−	−	−	−	0.52	−	2.48	−	−	−	1.10	0.35	1.94
Flavonols													
Myricetin	−	−	−	−	−	−	0.39	0.41	0.02	−	−	−	0.28
Rutin	−	−	−	0.02	−	0.01	−	−	−	0.03	−	−	0.02
Astragalin	−	−	−	0.01	−	−	−	0.01	−	0.01	−	−	−
Hyperoside	−	−	−	−	−	0.01	0.20	0.26	−	−	−	0.01	0.47
Galangin	1.21	0.88	0.60	0.46	0.36	0.20	0.24	0.19	0.18	−	−	−	−
Flavones													
Chrysin	0.60	−	−	−	−	−	−	−	−	−	−	−	−
Luteolin	−	−	−	−	−	−	−	0.10	−	−	−	−	−
Cynaroside	−	−	−	0.67	−	−	0.48	0.53	0.50	0.57	−	−	0.35
Apigetrin	−	−	−	0.01	−	−	0.01	0.01	0.01	0.01	−	−	0.01
Apigenin	0.06	−	0.03	−	−	−	−	−	−	−	−	−	−
Hydroxycoumarins													
Aesculin	−	−	0.05	−	−	−	0.65	0.69	−	−	−	0.44	0.87
Stilbenes													
Resveratrol	−	−	−	−	−	−	4.00	−	−	−	−	−	−
Flavanones													
Hesperetin	−	−	0.09	0.05	−	−	0.71	0.44	−	0.10	−	−	0.41
Dihydrochalcones													
Phlorizin	−	−	−	0.08	0.13	0.18	1.03	0.98	0.13	0.06	0.05	0.07	0.98
Anthocyanins (mg/L)													
Myrtillin	−	−	−	0.02	−	−	0.47	0.05	−	−	−	−	0.57
Malvin	−	−	−	0.01	−	−	0.04	−	−	−	−	−	0.06
Cyanidin 3-O-(2″-xylosyl)glucoside	−	−	−	−	−	−	0.04	0.02	−	−	−	−	0.04
Chrysanthemin	−	−	−	−	−	−	0.15	0.02	−	−	−	−	0.28
Peonidin 3-O-glucoside	−	−	−	0.09	−	−	0.61	0.06	−	−	−	−	0.69
Oenin	−	−	−	0.55	−	−	3.43	0.64	−	−	−	−	3.86
RSA (mmol TE/L)	0.39	0.44	0.42	1.97	3.17	2.68	16.09	18.71	1.86	1.64	1.26	0.82	12.03
TPC (g GAE/L)	1.15	1.44	1.41	1.44	1.51	1.67	2.24	2.36	1.35	1.31	0.81	1.17	1.71

For better visibility standard errors are given in Appendix A.

**Table 4 foods-10-00158-t004:** High resolution MS data of anthocyanins and related compounds identified in three Italian red and one rose wine samples. Peak No.—peak numbers; *Δ* ppm—mean mass accuracy; + stands for detected and − stands for not detected compound.

Peak No.	Anthocyanins	Molecular Formula, M^+^ (*m/z*)	Calculated Mass, M^+^ (*m/z*)	Exact Mass, M^+^ (*m/z*)	*Δ* ppm	W4	W7	W8	W13
1	Delphinidin 3-O-glucoside (Myrtillin) ^a,b^	C_21_H_21_O_12_^+^	465.10275	465.10123	3.27	+	+	+	+
2	Malvidin 3,5-di-O-glucoside (Malvin) ^a,b^	C_29_H_35_O_17_^+^	655.18688	655.18469	3.34	+	+	+	+
3	Cyanidin 3-O-(2″-xylosyl)glucoside ^a,b^	C_26_H_29_O_15_^+^	581.15010	581.14911	1.70	−	+	+	+
4	Cyanidin 3-O-glucoside (Chrysanthemin) ^a,b^	C_21_H_21_O_11_^+^	449.10784	449.10684	2.23	−	+	+	+
5	Delphinidin 3-O-glucoside-pyruvate ^d^	C_24_H_21_O_14_^+^	533.09258	533.09176	1.54	−	+	+	+
6	Petunidin 3-O-glucoside isomer 1 ^b^	C_22_H_23_O_12_^+^	479.11840	479.11710	2.71	+	+	+	+
7	Petunidin 3-O-glucoside-acetaldehyde ^d^	C_24_H_23_O_12_^+^	503.11840	503.11703	2.72	−	+	−	+
8	Peonidin 3-O-glucoside ^a,b^	C_22_H_23_O_11_^+^	463.12349	463.12231	2.55	+	+	+	+
9	Malvidin 3-O-glucoside (Oenin) ^a,b^	C_23_H_25_O_12_^+^	493.13405	493.13300	2.13	+	+	+	+
10	Delphinidin 3-O-(6″-acetyl)glucoside ^b^	C_23_H_23_O_13_^+^	507.11332	507.11215	2.31	+	+	+	+
11	Malvidin 3-O-glucoside-acetaldehyde ^d^	C_25_H_25_O_12_^+^	517.13405	517.13254	2.92	−	+	−	+
12	Peonidin 3-O-glucoside-pyruvate ^d^	C_25_H_25_O_13_^+^	531.11332	531.11145	3.52	+	+	+	+
13	Malvidin 3-O-glucoside-pyruvate ^d^	C_26_H_25_O_14_^+^	561.12388	561.12256	2.35	+	+	+	+
14	Petunidin 3-O-(6″-acetyl)glucoside ^b^	C_24_H_25_O_13_^+^	521.12897	521.12732	3.17	+	+	+	+
15	Malvidin 3-O-glucoside-acetone ^d^	C_26_H_27_O_12_^+^	531.14970	531.14862	2.03	−	+	+	+
16	Malvidin 3-O-(6″-acetyl)glucoside-acetaldehyde ^d^	C_27_H_27_O_13_^+^	559.14462	559.14362	1.79	+	+	+	+
17	Malvidin 3-O-glucoside-8-ethyl-(epi)catechin ^d^	C_40_H_41_O_18_^+^	809.22874	809.22797	0.95	−	+	+	+
18	Malvidin 3-O-(6″-acetyl)glucoside-pyruvate ^d^	C_28_H_27_O_15_^+^	603.13445	603.13293	2.52	−	+	+	+
19	Peonidin 3-O-(6″-acetyl)glucoside ^b^	C_24_H_25_O_12_^+^	505.13405	505.13358	0.93	+	+	+	+
20	Malvidin 3-O-(6″-acetyl)glucoside ^b^	C_25_H_27_O_13_^+^	535.14462	535.14325	2.56	+	+	+	+
21	Delphinidin 3-O-(6″-p-coumaroyl)glucoside ^b^	C_30_H_27_O_14_^+^	611.13953	611.13831	2.00	+	+	+	+
22	Petunidin 3-O-(6″-p-coumaroyl)glucoside-8-ethyl-(epi)catechin ^d^	C_48_H_45_O_20_^+^	941.24987	941.24832	1.65	−	+	−	+
23	Malvidin 3-O-glucoside-4-vinyl-(epi)catechin ^d^	C_40_H_37_O_18_^+^	805.19744	805.19623	1.50	−	+	+	+
24	Malvidin 3-O-(6″-p-coumaroyl)glucoside-acetaldehyde ^d^	C_34_H_31_O_14_^+^	663.17083	663.16943	2.11	−	+	−	+
25	Cyanidin 3-O-(6″-p-coumaroyl)glucoside ^b^	C_30_H_27_O_13_^+^	595.14462	595.14347	1.93	+	+	+	+
26	Malvidin 3-O-(6″-p-coumaroyl)glucoside-8-ethyl-(epi)catechin ^d^	C_49_H_47_O_20_^+^	955.26552	955.26459	0.97	−	+	+	+
27	Delphinidin 3-O-glucuronide ^b^	C_21_H_19_O_13_^+^	479.08202	479.08092	2.30	+	+	+	+
28	Malvidin 3-O-(6″-p-coumaroyl)glucoside-pyruvate ^d^	C_35_H_31_O_16_^+^	707.16066	707.15997	0.98	−	+	+	+
29	Malvidin 3-O-glucoside-4-vinylcatechol ^d^	C_31_H_29_O_14_^+^	625.15518	625.15369	2.38	+	+	+	+
30	Malvidin 3-O-(6″-p-coumaroyl)glucoside ^b^	C_32_H_31_O_14_^+^	639.17083	639.17004	1.24	−	+	+	+
31	Peonidin 3-O-(6″-p-coumaroyl)glucoside ^b^	C_31_H_29_O_13_^+^	609.16027	609.15955	1.18	−	+	−	+
32	Peonidin 3-O-glucoside-4-vinylphenol ^d^	C_30_H_27_O_12_^+^	579.14970	579.14868	1.76	+	+	+	+
33	Malvidin 3-O-glucoside-4-vinylphenol ^d^	C_31_H_29_O_13_^+^	609.16027	609.15857	2.79	+	+	+	+
34	Petunidin 3-O-glucoside isomer 2 ^b^	C_22_H_23_O_12_^+^	479.11840	479.11731	2.28	+	−	−	−
35	Malvidin 3-O-glucoside-4-vinylguaiacol ^d^	C_32_H_31_O_14_^+^	639.17083	639.17004	1.24	+	+	+	+
36	Malvidin 3-O-glucoside-pyranone ^d^	C_25_H_25_O_13_^+^	533.12897	533.12817	1.50	−	+	+	+
37	Malvidin 3-O-(6″-acetyl)glucoside-4-vinylphenol ^d^	C_33_H_31_O_14_^+^	651.17083	651.17041	0.64	+	+	+	+
38	Malvidin 3-O-(6″-p-coumaroyl)glucoside-4-vinylcatechol ^d^	C_40_H_35_O_16_^+^	771.19196	771.19006	2.46	+	+	+	+
39	Malvidin-pyruvate ^d^	C_20_H_15_O_9_^+^	399.07106	399.06982	3.11	+	+	+	+
40	Malvidin 3-O-(6″-p-coumaroyl)glucoside-4-vinylphenol ^d^	C_40_H_35_O_15_^+^	755.19705	755.19647	0.77	+	+	+	+
41	Delphinidin ^c^	C_15_H_11_O_7_^+^	303.04993	303.04916	2.54	+	+	+	+
42	Petunidin ^c^	C_16_H_13_O_7_^+^	317.06558	317.06418	4.42	−	+	+	+

^a^ Confirmed using standards; ^b^ Anthocyanins; ^c^ Anthocyanidins; ^d^ Pyroanthocyanins.

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
