# Peer review of "Elemental Analysis and Phenolic Profiles of Selected Italian Wines"

_foods, 2021, doi:10.3390/foods10010158_

Round 1
Reviewer 1 Report
General comments
This paper deals with the characterization of some Italian wines from elemental composition and polyphenolic profiles. In my opinion, the topic fits the scope of Foods but I have serious concerns about its quality. Some overall comments to be addressed are as follows.
The number of samples under study is certainly limited so that conclusions will be merely speculative. There are not several wines with the same characteristics so we cannot have an idea about variability ranges concerning each given class.
In the case of antioxidant indexes, authors state that the analysis was carried out in triplicate. It is not clear for the other methods, but I have doubts about this. Without replicates, the possibilities of systematic errors increase and the influence of the experimental variability on the PCA results cannot be evaluated. This also hinders the extraction of reliable conclusions.
The introduction should be significantly improved with a more comprehensive revision of the topic, focusing on wine classification and authentication based on elemental and phenolic composition (at least). Besides, references should be better selected. For instance, authors write “Chemical characterization of wine represents necessary step in wine authenticity examinations”. There are numerous references addressing this topic, some of them reviews that will cover the topic form a broader point of view. This comment can be extended to the entire introduction.
Data from phenolic acid concentrations should also be treated by PCA in order to try to find class descriptors. In the literature, polyphenols have been extensively used for characterization and authentication purposes. This could contribute to enrich the manuscript.
The absence of QCs to evaluate the quality of the data and the PCA model is another hindrance.
Other specific points are given as follows:
Title: it should be more informative.
Abstract:
Be consistent with the verbal tenses.
The abstract is not well organized. For polyphenols, for instance, explain first the methods and then comment on the range of compounds to be determined/identified.
How many metals were quantified? Please add.
According to the lines 19 to 21, it is unclear how many anthocyanins were identified. Besides, why anthocyanins are highlighted? since they are just a subfamily of phenolic compounds. Why the other subfamilies have not been mentioned?
Line 24: “Total phenolic content (TPC) and radical scavenging activity (RSA) were also measured”. Mention the methods for these indexes.
Materials and methods:
Materials and methods: Are 13 samples representative to provide solid conclusions on the behavior of wine?
Line 102. The list of polyphenols should be given.
Line 112. Thermo scientific, please correct.
Line 115. Indicate which elements.
Line 177. Extracts of what?? NO extraction procedure has been reported. I thought wines were just filtered prior analysis. Something missing?
SPE cartridges are mention but no SPE procedure is detailed.
Why different gradients were applied to the two LC-MS? Please explain.
Data Analysis: data preprocessing should be indicated.
A quality control sample (mixture of equal volumes from all samples) should be prepared and analyzed. QCs are normally analyzed inserted between the samples (e.g. every 5 or 10 samples) to evaluate the repeatability of the analytical methods as well as significance of the chemometric results.
Results and discussion
Figure 1. As far as I see, sample types are not well separated, and wines from the different origins spread throughout the score plot without clearly discriminated groups. If so, the information gained from elemental composition is quite irrelevant, meaning the figure can be suppressed.
Figure 2. Please combine info from a and b in a single plot. Besides add the plot of loadings from a similar view angle for a better interpretation of the descriptive ability of elements.
Figures 3, 4 and 5 are scarcely informative and quite repetitive. If you want to compare classes according to the elemental composition, I suggest box plots with whiskers on relevant examples (i.e. facing the most discriminant elements).
Line 313. Please rephrase.
Regarding to phenolic indexes, again boxplots could be useful to better describe the behavior of the samples.
Any conclusion from both FC and DPPH data? as they both express parameters related to the antioxidant activity of wines.
Line 343. “A total of twenty-eight phenolic compounds and seven anthocyanins”. Anthocyanins are also phenolic compounds.
Line 344 and table 3. Authors provide a list of compounds “Among the identified compounds, gallic, ellagic, gentisic and protocatechuic acids belong to hydroxybenzoic acids; chlorogenic, caffeic, p-coumaric…” Other compounds commonly found in wines, some of them quite relevant (caftaric acid, coutaric acid, vanillic acid, ethyl gallate, epicatechin, etc.), have not been investigated. Authors should discuss why some compounds have been selected while other have been dismissed.
Authors have worked on the tentative identification of anthocyanins by LC-HRMS. A discussion on the main objective of that study should be included. Besides, it is not clear why the study was only focused on anthocyanins and not in other families.
Author Response
Reviewer 1
General comments
This paper deals with the characterization of some Italian wines from elemental composition and polyphenolic profiles. In my opinion, the topic fits the scope of Foods but I have serious concerns about its quality. Some overall comments to be addressed are as follows.
We would like to emphasize that the study of the chemical composition of wines is nowadays a topic of great interest because of the importance of this market, especially in Italy, and also considering the numerous cases of falsification of famous and very expensive wines.
Having new data in this field can only lead to an improvement of knowledge and moreover, to our knowledge, it has never been published a work comparing these 3 regions where, we want to underline it, the wines come from the same winery.
The number of samples under study is certainly limited so that conclusions will be merely speculative. There are not several wines with the same characteristics so we cannot have an idea about variability ranges concerning each given class.
We agree with the referee that the number of samples is limited but they have the particularity of belonging to the same winery and for this reason it was interesting to compare wines coming from different regions but produced from vines managed by the same winery.
In the case of antioxidant indexes, authors state that the analysis was carried out in triplicate. It is not clear for the other methods, but I have doubts about this. Without replicates, the possibilities of systematic errors increase and the influence of the experimental variability on the PCA results cannot be evaluated. This also hinders the extraction of reliable conclusions.
At this purpose it is not clear what the reviewers intends with analyses carried out in triplicate. Obviously ICP-OES and ICP-MS analysis are carried out in triplicate and for each element RSD% is furnished by the instrument. In the supplementary material a table with the standard deviation for each element has now been included. Being low the errors associated to the determination of all the analyzed elements, this doesn’t compromise in any way the validity of the results obtained from PCA analysis. PCA has been carried out by Statistica package software, as specified in the materials and methods sections. The errors on the variables are not required as input but is obvious that this represents the base for the goodness of the results obtainable. Tables with standard deviations have been inserted in the supplementary material.
The introduction should be significantly improved with a more comprehensive revision of the topic, focusing on wine classification and authentication based on elemental and phenolic composition (at least). Besides, references should be better selected. For instance, authors write “Chemical characterization of wine represents necessary step in wine authenticity examinations”. There are numerous references addressing this topic, some of them reviews that will cover the topic form a broader point of view. This comment can be extended to the entire introduction.
I would answer in this way: The references reported are quite comprehensive of the entire topic treated; in fact, what the reviewer requires is already present in the introduction. We would like to underline that the present paper is not a review on the topic but the aim is to acquire some more data on specific groups of wines.
Data from phenolic acid concentrations should also be treated by PCA in order to try to find class descriptors. In the literature, polyphenols have been extensively used for characterization and authentication purposes. This could contribute to enrich the manuscript.
This has been tried and the following figure included in the text; what is more evident is the group formed by the red wines. The comment has been also introduced in the text.
The absence of QCs to evaluate the quality of the data and the PCA model is another hindrance.
The instruments used for performing all the analyses were under quality controls and from this comes the quality of the data obtained from PCA as just previously mentioned. At this purpose Tables S2 and S3 have been added reporting standard deviations for the data acquired.
Other specific points are given as follows:
Title: it should be more informative.
We have modified in this way in accordance also to what suggested by reviewer 3: Elemental analysis and phenolic profiles of selected Italian wines
Abstract:
Be consistent with the verbal tenses. ?? we have checked the English form
The abstract is not well organized. For polyphenols, for instance, explain first the methods and then comment on the range of compounds to be determined/identified.
We have checked again the abstract and the order seems the same, i.e. metals and method of analysis and polyphenols and methods of analyses.
How many metals were quantified? Please add. 20 metals were quantified; it has been added.
According to the lines 19 to 21, it is unclear how many anthocyanins were identified. Besides, why anthocyanins are highlighted? Since they are just a subfamily of phenolic compounds. Why the other subfamilies have not been mentioned?
Thank you. It was not clear, but now it is corrected in the new version of the revised manuscript. There are sentences in the abstract: "42 anthocyanins were identified (among these 6 were those also quantified)". To avoid confusion, the compounds tested in this work were labeled with non-anthocyanins and anthocyanins. We think that now everything is clearly explained throughout the entire manuscript.
We agree that anthocyanins are a subfamily of phenolic compounds. In this paper, we analyzed phenolic acids, flavonoids and their derivatives in the negative ionization mode, while anthocyanins were analyzed in the positive ionization mode. In the absence of standards, we decided to identify a larger number of anthocyanins using advanced high-resolution mass spectrometry combined with multi-stage mass spectrometry. This is why anthocyanins are prominent.
Other groups of phenolic compounds have not been the subject of research in this paper, but we will investigate them in the future.
Line 24: “Total phenolic content (TPC) and radical scavenging activity (RSA) were also measured”. Mention the methods for these indexes. It is included.
Materials and methods:
Materials and methods: Are 13 samples representative to provide solid conclusions on the behavior of wine? It is worth noting that the wines analyzed in this paper come from the same winemaker (Sagrivit) which is a very important wine maker on the Italian market. More details on this company have been added in the text. We are aware that the data set could be improved but for the moment we decided to analyze this first selection of wines provided by Sagrivit. The results obtained are very encouraging and surely in the future we will expand the number.
Line 102. The list of polyphenols should be given.
The list is given in the Results and Discussion part. And also in the Table 3.
Line 112. Thermo scientific, please correct.
Corrected.
Line 115. Indicate which elements.
The list is given in the Table 2
Line 177. Extracts of what?? NO extraction procedure has been reported. I thought wines were just filtered prior analysis. Something missing?
Corrected.
SPE cartridges are mention but no SPE procedure is detailed.
Detail of SPE method is given in the 2.4.3. UHPLC-LTQ OrbiTrap MS analysis of anthocyanins
Why different gradients were applied to the two LC-MS? Please explain.
We used different mass detectors for two LC/MS analyzes. They are developed independently of each other and are very similar. According to the editor's recommendation, this part of the text has been shortened, citing the appropriate literature.
Data Analysis: data preprocessing should be indicated.
A quality control sample (mixture of equal volumes from all samples) should be prepared and analyzed. QCs are normally analyzed inserted between the samples (e.g. every 5 or 10 samples) to evaluate the repeatability of the analytical methods as well as significance of the chemometric results.
We are fully agreed with this comment, but we are not able to provide these results now. We did not approach the experiment in that way, and now, due to the lack of some samples and a pandemic, we cannot do the necessary experiments.
Results and discussion
Figure 1. As far as I see, sample types are not well separated, and wines from the different origins spread throughout the score plot without clearly discriminated groups. If so, the information gained from elemental composition is quite irrelevant, meaning the figure can be suppressed.
Ok, it has been moved to supplementary materials; it cannot be delated since it is important to suggest which variables were eliminated before running again the calculation by PCA.
Figure 2. Please combine info from a and b in a single plot. Besides add the plot of loadings from a similar view angle for a better interpretation of the descriptive ability of elements.
Unfortunately, it is not possible to combine them in a single plot otherwise the figures are not readable since labels, marks and circles would be superimposed. The loading plots, to be readable, should be plotted in two dimensions (so in the supplementary there are 2 plots: PC1/PC3 and PC1/PC2, see figure S3).
Figures 3, 4 and 5 are scarcely informative and quite repetitive. If you want to compare classes according to the elemental composition, I suggest box plots with whiskers on relevant examples (i.e. facing the most discriminant elements).
If you pay attention at these figures it is evident how they allow to highlight some compositional differences among the wines coming from the 3 regions. The trends, especially as far as main elements are considered, seem similar but really there are some significant differences especially in the intensities.
We have also tried the representation suggested by the reviewer, i.e. box and whiskers plots, and the figures obtained are reported below. It is worth noting that what observed in figures 2, 3 and 4 (new numbers of ex figures 3,4 and 5 in the previous version of the paper) is basically confirmed by these new graphs “box and whiskers” and any additional information is added with the exception that it is more evident that Friuli and Veneto wines show a slightly higher concentrations of Sb and Ni with respect to Umbria wines (this comment has been added in the text). Therefore, box and whiskers graphs (due to the high number of figures of the paper and due to the fact that they don’t bring more significant information) have been added in the supplementary materials (see also supplementary materials for the captions).
Figure S4
Figure S5
Figure S6
Line 313. Please rephrase.
Corrected.
Regarding to phenolic indexes, again boxplots could be useful to better describe the behavior of the samples. Any conclusion from both FC and DPPH data? as they both express parameters related to the antioxidant activity of wines.
I think that the most useful representation is as a graph:
From where it can be observed the most evident difference are RSA values which are higher for red wines, as already commented in the text (we think that due to the high number of figures already present this graph is not necessary).
Line 343. “A total of twenty-eight phenolic compounds and seven anthocyanins”. Anthocyanins are also phenolic compounds.
This is corrected. See some previous answer about this comment.
Line 344 and table 3. Authors provide a list of compounds “Among the identified compounds, gallic, ellagic, gentisic and protocatechuic acids belong to hydroxybenzoic acids; chlorogenic, caffeic, p-coumaric…” Other compounds commonly found in wines, some of them quite relevant (caftaric acid, coutaric acid, vanillic acid, ethyl gallate, epicatechin, etc.), have not been investigated. Authors should discuss why some compounds have been selected while other have been dismissed.
Simply, we tested for the presence of all the phenolic compounds we had in the lab as standards. There are a large number of compounds in wine that are not available as standards, so they cannot be quantified. In further research of wine and grapes, we plan to develop more standards that are specific to wine and grapes.
Authors have worked on the tentative identification of anthocyanins by LC-HRMS. A discussion on the main objective of that study should be included. Besides, it is not clear why the study was only focused on anthocyanins and not in other families.
In this paper, we run HRMS analyzes in the positive ionization mode. And under such conditions, we got the greatest detector responses for anthocyanins. Some other compounds that are not the subject of this paper could be analyzed with different experimental conditions. We had only six standards of anthocyanins, and we wanted to try to identify as many compounds as possible. In the absence of standards, a larger number of anthocyanins were tentatively identified using advanced high-resolution mass spectrometry combined with multi stage mass spectrometry.
Reviewer 2 Report
The paper deals with the study of 13 Italian wines from different origins regarding metals and phenolic compounds. The work is relevant, it is well presented, however there are some points that need to be clarified. For instance, in the first part of the paper authors refer to anthocyanin-derivatives as anthocyanins. That should be changed in the manuscript during the discussion.
Lines 19-21: After the second sentence of the abstract, it seems that phenolic compounds were identified by ICP-OES and ICP-MS when you say “A total of twenty-eight phenolic compounds and six anthocyanins were identified and quantified. Furthermore in 4 samples (1 rosè and 3 red wines) 42 anthocyanins were identified.” You should begin the sentence “Regarding phenolic compounds, a total of twenty-eight compounds and…” In addition, when you say that 42 anthocyanins were identified in 4 samples, do you mean 42 anthocyanin-derived compounds? The anthocyanins present in wines are usually 15 (5 non-acylated, 5 acetic acid derivatives and 5 coumaric acid derivatives), you can also have some caffeoyl derivatives but it is impossible to have 42 different anthocyanins in wines.
Lines 29-30: “Most of the anthocyanins identified are formed during the ageing of wine.” This sentence is no accurate as anthocyanins are naturally present in grapes and their existence in wines is due to their extraction during the winemaking process. During wine ageing anthocyanins concentration decrease dramatically due to degradation, polymerization and reaction with other wine components yielding to the formation of anthocyanin-derived compounds. In this sentence I think you are referring to anthocyanin-derived compounds and not anthocyanins and by this the sentence should be changed.
Lines 46-47: I think you should remove the last part of the sentence “therefore wines are nowadays considered functional food”. Due to the high alcoholic content of wines that is responsible for numerous diseases and deaths it is not responsible to postulate wine as a functional food.
Line 84: “Friuli” is in a different font (size).
Line 98: correct to “anthocyanins”
Line 112: the word “Scientific” should corrected
Lines 118, 1234-124: Which phenolic compounds standards were used for the calibration curves? And anthocyanins?
Line 148: Do you mean anthocyanin-derivatives
Results and discussion: I think you should make an introduction to the main goals of the work and what you will do to achieve them.
Lines 206-207: And how about Italian wines? How are the levels of magnesium and calcium?
It should be noticed that I don’t have expertise on the PCA analysis.
In Table S2, you use several times the designation of “hexoside” but why did you not use glucose as the anthocyanins present in grapes and wines are always glucose derivatives. “Delphinidin 3-O-hexoside-pyruvate” is carboxypyranodelphinidin 3-O-glucoside” or “Delphinidin 3-O-hexoside-pyruvate adduct” or the delphinidin derivative of A-type vitisin. The same for the other anthocyanin derivatives. Moreover, “Petunidin 3-O-hexoside-acetaldehyde” is “Pyranopetunidin 3-O-glucoside” or the petunidin derivative of B-type vitisin. “Malvidin 3-O-hexoside-acetone” is the “Methylpyranomalvidin-3-O-glucoside”.
In the discussion you also refer to hexoside derivates when you can use the term glucoside.
Line 401: anthocyanidin
Line 404: during wine ageing
Table 3: Why didn’t you quantify the anthocyanins?
In Table 4 or Table S2 you should add references for the anthocyanin-derived compounds. This compounds are already described in the literature.
In Table 4 it would be important if you can include if some of the compounds are present in. higher amounts in the different wines.
You have the references duplicated. They appear in numbers but then appear as author and year. Please change.
Author Response
Review 2
The paper deals with the study of 13 Italian wines from different origins regarding metals and phenolic compounds. The work is relevant, it is well presented, however there are some points that need to be clarified. For instance, in the first part of the paper authors refer to anthocyanin-derivatives as anthocyanins. That should be changed in the manuscript during the discussion.
We have clarified this point in the text.
Lines 19-21: After the second sentence of the abstract, it seems that phenolic compounds were identified by ICP-OES and ICP-MS when you say “A total of twenty-eight phenolic compounds and six anthocyanins were identified and quantified. Furthermore in 4 samples (1 rosè and 3 red wines) 42 anthocyanins were identified.” You should begin the sentence “Regarding phenolic compounds, a total of twenty-eight compounds and…” In addition, when you say that 42 anthocyanins were identified in 4 samples, do you mean 42 anthocyanin-derived compounds? The anthocyanins present in wines are usually 15 (5 non-acylated, 5 acetic acid derivatives and 5 coumaric acid derivatives), you can also have some caffeoyl derivatives but it is impossible to have 42 different anthocyanins in wines.
Lines 29-30: “Most of the anthocyanins identified are formed during the ageing of wine.” This sentence is no accurate as anthocyanins are naturally present in grapes and their existence in wines is due to their extraction during the winemaking process. During wine ageing anthocyanins concentration decrease dramatically due to degradation, polymerization and reaction with other wine components yielding to the formation of anthocyanin-derived compounds. In this sentence I think you are referring to anthocyanin-derived compounds and not anthocyanins and by this the sentence should be changed.
The entire abstract has now been modified and corrected accordingly. By anthocyanins derivatives, we mean all compounds in Table 4 (including pyranoanthocyanins)
Lines 46-47: I think you should remove the last part of the sentence “therefore wines are nowadays considered functional food”. Due to the high alcoholic content of wines that is responsible for numerous diseases and deaths it is not responsible to postulate wine as a functional food.
Really wine has positive effects on human health (at this purpose see what introduced now in the new version of the paper at lines 69-72)
Line 84: “Friuli” is in a different font (size). corrected
Line 98: correct to “anthocyanins” corrected
Line 112: the word “Scientific” should corrected
Lines 118, 1234-124: Which phenolic compounds standards were used for the calibration curves? And anthocyanins?
All compounds listed in Table 3 were used to construct the calibration curve. It is a large number of compounds, so they are not listed here, but only in Table 3.
Line 148: Do you mean anthocyanin-derivatives
No, this section deals with the analysis of non-anthocyanins. The analysis of anthocyanins is given in the next paragraph.
Results and discussion: I think you should make an introduction to the main goals of the work and what you will do to achieve them.
The following sentence has been introduced: The main goals of the present study are to achieve information on the specific characteristic of the analyzed wines trying to highlight the differences through the combined application of analytical methodologies and data treatment by statistical methods.
Lines 206-207: And how about Italian wines? How are the levels of magnesium and calcium?
It is reported at line 323 of the new version of the text.
It should be noticed that I don’t have expertise on the PCA analysis.
In Table S2, you use several times the designation of “hexoside” but why did you not use glucose as the anthocyanins present in grapes and wines are always glucose derivatives. “Delphinidin 3-O-hexoside-pyruvate” is carboxypyranodelphinidin 3-O-glucoside” or “Delphinidin 3-O-hexoside-pyruvate adduct” or the delphinidin derivative of A-type vitisin. The same for the other anthocyanin derivatives. Moreover, “Petunidin 3-O-hexoside-acetaldehyde” is “Pyranopetunidin 3-O-glucoside” or the petunidin derivative of B-type vitisin. “Malvidin 3-O-hexoside-acetone” is the “Methylpyranomalvidin-3-O-glucoside”.
We agree with you regarding all comments. But still, we decided to be a little more careful. Through this LC / HRMS technique, we cannot know exactly whether they are glucose or galactose derivatives. Of course, the cornerstone is glucose, but we fenced ourselves off this way by writing hexose. As for naming, we used information from the literature.
In the discussion you also refer to hexoside derivates when you can use the term glucoside.
See previous comment.
Line 401: anthocyanidin
It refer to anthocyanin aglycones. Sentence are corrected.
Line 404: during wine ageing
We are using words specific to US grammar.
Table 3: Why didn’t you quantify the anthocyanins?
We only quantified six of them, because we have these six standards. See end of Table 3.
In Table 4 or Table S2 you should add references for the anthocyanin-derived compounds. These compounds are already described in the literature.
Yes, all of these compounds are previously identified in wine. We add our reference about it in the text.
In Table 4 it would be important if you can include if some of the compounds are present in. higher amounts in the different wines.
The only thing we could compare are the concentrations, because the high value of peak area does not say that a given compound is the most represented, but says that under the given experimental conditions the response of the detector for that compound is the largest. This can be due to the stability of the molecular ion, the type of ionization, the polarity. We only had 6 standards available of antocyanins. As for the differences between the samples, at this time, we can only show + and -, because the data of abundance of the compounds are not currently available to us due to the corona virus pandemic.
You have the references duplicated. They appear in numbers but then appear as author and year. Please change. We have corrected
Reviewer 3 Report
Manuscript ID: foods-1050011
Title: Elemental analysis and phenolic profiles of Italian wines coming from different regions.
This study is a good descriptive work of wines coming from different Italian regions. Moreover, samples show other interesting differences in agronomical parameters. The authors report a wide range of parameters such as elemental and phenolic composition. Perhaps this type of study can be accused of lack of novelty. However, I believe that due to the high variability and changing conditions in the wine industry, this argument does not make sense if the study is well planned. Even more so in the current global climate situation.
For me, the biggest concern about the study is that although authors continuously compare parameters obtained for the different wines according their region, variety, type of wine or type of soil, they did not develop any analysis of variance (ANOVA). For me, ANOVAs are mandatory for this study be published in Foods. Even if non-significant differences are found, this result would be also interesting and would have scientific value. The ANOVA would make it possible to compare the averages of the different parameters analysed and the analysis of contrasts, such as the Tuckey test, will make it possible to identify which groups are or are not significantly different.
Title
How important is ‘coming from different regions’ in the title? A very attractive title will be get if this is remove.
Abstract
Abstract is well structured. It described correctly the methods and results of the work. Perhaps, a couple of introductory sentences and the most important quantitative results might be added.
Lines 20-21: Six or 42 anthocyanins. In the manuscript is well explained, but in the abstract it is a little hard of understand.
Introduction
Introduction section is concise and correctly explain the background of the study.
Line 36: Better express the production in t or millions of t. Mt can me ambiguous.
Line 39: I think: … represents a necessary step…
Lines 40-47: Recently, nutritionists are reporting that the demonstrated positive effects of wine polyphenols on human health can be offset by the alcohol in the drink. Perhaps this should be added to the manuscript, or better, a reference demonstrating the positive effect of wine polyphenols even taking into account the alcohol content.
Line 52: I think flavanols (and not flavonols) are the main responsible for the color of white wines. Please, check it.
Line 62: As you described above, differences are not only due to the geographical area of origin. Better ‘… as affected by the abovementioned factors’.
Line 67: This sentence is a bit confusing; I would split it in two: ‘…on which the vines are grown. From the grape, metals can reach…’.
Material and Methods
M&M is a very complete section. The authors developed a comprehensive study on the chemical composition of wine samples related to the health properties of these products. Only a more detailed description of two of the methods is needed.
Line 86: What ‘which is one of the largest national agricultural realities’ means? It is not clear to me.
Line 107: As you included a reactive section, you can avoid the information in brackets in the following.
Line 112: There is a strange symbol in the word ‘Scientic’ in my version.
Lines 110-115: These methods have to be described with more detail or references have to be added.
Lines 133-134: Please, be consistent in the way you express the different steps in the gradient elution.
Line 183. There is a missing full-stop.
Results and discussion
This section is, in general well, addressed. However, I strongly suggest the use of ANOVAs for looking for significant differences between the different groups of samples and analysis of contrasts for identifying which groups are or are not different.
Line 205: This sentence is a bit confusing.
Lines 207-208: Have you developed an Analysis of Variance (ANOVA). If not, it is not adequate to use this expression. Moreover, it would be really interesting to develop different ANOVAs to check if there are significant differences between regions or between different wines in a same region. Or even between your results and results of other authors.
Line 211: For example, is it this difference statistically significant? Adding ANOVAs and analysis of contrasts, such as Tuckey test, will increase a lot its scientific value of the study.
Line 218: Please, moderate this type of expressions. Although the number of wines taken in to account in your study is adequate, I think it is not enough to say ‘Italian wines’. Better say ‘The Italian wines taken in to account in this study’ or ‘The studied Italian wines’.
Table 2: You did each measurement in triplicate so, please add the standard errors of the means reported. Perhaps in a supplementary table.
Figure 2 caption: Please describe the data shown in figures 2a and 2b.
Lines 264-268: If you want to highlight these relationships between samples of the same variety in a same region you can measure the distance between sample in the PCA. You can use Euclidean or Mahalanobis distance.
Lines 272-279: For me, an ANOVA would be really useful for explain this. Even if there are no significant differences, it would be interesting to know.
Line 352: I think there is a problem with the cites here and in the remaining manuscript. ¿Why is the author name added to each cite?
Table 3: Again, add the standard errors.
Author Response
Reviewer 3
Title: Elemental analysis and phenolic profiles of Italian wines coming from different regions.
This study is a good descriptive work of wines coming from different Italian regions. Moreover, samples show other interesting differences in agronomical parameters. The authors report a wide range of parameters such as elemental and phenolic composition. Perhaps this type of study can be accused of lack of novelty.
In our opinion the work cannot be accused of lake of novelty because, although analytical methods are known and conventionally used for this kind of analysis, the novelty is represented by the comparison between wines coming from the same winemaker (Sagrivit) from 3 specific Italian regions and this comparison, to our knowledge, has never been reported in literature. Sagrivit is a very important wine maker on the Italian market. More details on this company have been added in the text. We are aware that the data set could be improved but for the moment we decided to analyze this first selection of wines provided by Sagrivit. The results obtained are very encouraging and surely in the future we will expand the number of wines.
However, I believe that due to the high variability and changing conditions in the wine industry, this argument does not make sense if the study is well planned. Even more so in the current global climate situation.
In fact, we agree with the reviewer and we have tried to put a lot of information in the work. The wine activity in Italy is one of the most developed in the world and for this reason a study such as ours study can only bring a further contribution to the wine market.
For me, the biggest concern about the study is that although authors continuously compare parameters obtained for the different wines according their region, variety, type of wine or type of soil, they did not develop any analysis of variance (ANOVA). For me, ANOVAs are mandatory for this study be published in Foods. Even if non-significant differences are found, this result would be also interesting and would have scientific value.
The ANOVA would make it possible to compare the averages of the different parameters analysed and the analysis of contrasts, such as the Tuckey test, will make it possible to identify which groups are or are not significantly different.
The authors are grateful to the reviewer suggestion and have tried to perform ANOVA one-way on both elemental data and phenolic composition. Nevertheless, the information obtained are not particularly significant and, in our opinion don’t add any fundamental information form the point of view of the discrimination among wines or description of their properties.
In our case we didn’t have enough data such as for example those reported in Novikova et al. 2020 where Tuckey test has been successfully applied. We have mentioned this in the text of the paper an added this reference:
Liubov Yu. Novikova and Lyudmila G. Naumova, Dependence of Fresh Grapes and Wine Taste Scores on the Origin of Varieties and Weather Conditions ofthe Harvest Year in the Northern Zone of Industrial Viticulture in Russia Agronomy 2020, 10, 1613; doi:10.3390/agronomy10101613 www.mdpi
At line 241 the following sentence has been introduced:
In order to get information on the origin of wine in terms of geographical provenance principal component analysis (PCA) were successfully applied while to highlight how specific condition (weather or harvest year) influence some wine properties such as the taste, other statistical methodologies including ANOVA and the Tuckey test were employed (ref. Nuvikova 2020)
Just to give the reviewer an idea on how we decided to proceed as far as elemental composition, at first, we have tried to divide wines according to the provenance/region and then we applied ANOVA one-way. Considering all the elements (20) any significant difference among groups appears with p=0.05 (F=0.12; Fcrit =3,158). Subsequently the wines were dived accordingly to the type (sparkling, white and red) but also in this case considering all the elements any significant difference was evidenced. In our opinion this approach applied to the data set considered is not able to disentangle any useful information. Surely, as underlined by the reviewer, there are other studies among those reported in the literature where ANOVA analysis was useful and applied for example for a preliminary selection of the variables.
Nevertheless, if we applied F test to couples the test is passed with F>Fcrit as shown in the results here reported:
test F |
VENETO/FRIULI |
||
|
Veneto |
Friuli |
|
Media |
140,0475 |
172,2363 |
|
Varianza |
60663,43449 |
114899,152 |
|
Osservazioni |
20 |
20 |
|
gdl |
19 |
19 |
|
F |
0,527971124 |
ok |
|
P(F<=f) una coda |
0,086505279 |
||
F crtitico una coda |
0,461201089 |
|
|
|
|||
test F |
VENETO/UMBIRA |
||
|
Veneto |
Umbria |
|
Media |
140,0475 |
187,8851 |
|
Varianza |
60663,43449 |
122991,279 |
|
Osservazioni |
20 |
20 |
|
gdl |
19 |
19 |
|
F |
0,493233627 |
ok |
|
P(F<=f) una coda |
0,066153789 |
||
F crtitico una coda |
0,461201089 |
|
|
|
|
|
|
test F |
FRIULI/UMBIRA |
||
Test F a due campioni per varianze |
|||
|
Friuli |
Umbria |
|
Media |
172,2363 |
187,8851 |
|
Varianza |
114899,1521 |
122991,279 |
|
Osservazioni |
20 |
20 |
|
gdl |
19 |
19 |
|
F |
0,934205687 |
||
P(F<=f) una coda |
0,441811341 |
||
F crtitico una coda |
0,461201089 |
|
|
An example of calculation of the variance associate to each variable and obtained after the Anova one-way is here reported:
In this case the difference is significant (F>Fcrit)-
On the contrary, the other statistical approach applied in the present study, such as PCA, i. e. a multivariate approach (ANOVA is univariate) allowed putting in evidence quite interesting information.
We have also searched in the literature (for example Food Chemistry 141 (2013) 4026–4030, Shen et al.) and ANOVA is in some cases couple to PCA. Nevertheless, there are numerous studied among those already reported in the references of our paper where ANOVA analysis is not applied.
It is worth noting that the variables selection to perform PCA analysis has allowed in our case to obtain a good separation among the groups.
Furthermore, ANOVA one-way was also applied to the analysis of phenols data.
At this purpose we have plotted all the phenols concentration for the 3 categories sparkling, white and red wines obtaining the following graph (that has been introduced in the text):
On the base of this graph, it is worth noting that the more significant differences among the 3 categories are the first 16 phenols.
ANOVA one-way has been therefore applied on the data set formed by the 3 categories and the first 16 phenols obtaining a significant difference among the 3 categories (p=0.15).
Title
How important is ‘coming from different regions’ in the title? A very attractive title will be get if this is remove.
Ok, we have removed it; the new title is: Elemental analysis and phenolic profiles of selected Italian wines
Abstract
Abstract is well structured. It described correctly the methods and results of the work. Perhaps, a couple of introductory sentences and the most important quantitative results might be added.
This sentence has been added: The study of the chemical composition of wines is nowadays a topic of great interest because of the importance of this market, especially in Italy, and also considering the numerous cases of falsification of famous and very expensive wines.
The most important results were already mentioned; the abstract has a limit of 250 words so we cannot add more
Lines 20-21: Six or 42 anthocyanins. In the manuscript is well explained, but in the abstract it is a little hard of understand.
It has been clarified adding this in the abstract: (among these 6 were those also quantified).
Introduction
Introduction section is concise and correctly explain the background of the study.
Line 36: Better express the production in t or millions of t. Mt can me ambiguous. corrected
Line 39: I think: … represents a necessary step… corrected
Lines 40-47: Recently, nutritionists are reporting that the demonstrated positive effects of wine polyphenols on human health can be offset by the alcohol in the drink. Perhaps this should be added to the manuscript, or better, a reference demonstrating the positive effect of wine polyphenols even taking into account the alcohol content.
References and remarks regarding positive effect of wine polyphenols in spite of alcohol content, has been added to the text.
Line 52: I think flavanols (and not flavonols) are the main responsible for the color of white wines. Please, check it.
Data available in the literature indicate flavonols as polyphenols mostly responsible for the color of white wines. “Regarding the colour of the wines, flavonols are yellow pigments that contribute directly to the colour of white wines” (Castillo-Muñoz, N., Gómez-Alonso, S., García-Romero, E., Hermosín-Gutiérrez, I. Flavonol profiles of Vitis vinifera white grape cultivars. J. Food Compos. Anal., 2010, 23(7), 699-705). “Grape skin is a source of natural pigments (anthocyanins and flavonols)… Flavonols constitute a group of flavonoids that vary in color from white to yellow and are closely related in structure to the flavones…“ (Benmezane, F., Cadot, Y., Djama, R., Djermoun, L. Determination of major anthocyanin pigments and flavonols in red grape skin of some table grape varieties (Vitis vinifera sp.) by highperformance liquid chromatography– photodiode array detection (HPLC-DAD). OENO One, 2016, 50(3), 125-135)
Line 62: As you described above, differences are not only due to the geographical area of origin. Better ‘… as affected by the abovementioned factors’. corrected
Line 67: This sentence is a bit confusing; I would split it in two: ‘…on which the vines are grown. From the grape, metals can reach…’. corrected
Material and Methods
M&M is a very complete section. The authors developed a comprehensive study on the chemical composition of wine samples related to the health properties of these products. Only a more detailed description of two of the methods is needed.
Line 86: What ‘which is one of the largest national agricultural realities’ means? It is not clear to me.
Sagrivit is one of Italy's largest agricultural companies. The company manages 14 historical companies, from the north to the south of the peninsula. Sagrivit operates in several sectors: cultivation of cereals, fruits, tobacco, cattle breeding, as well as four wine estates specialized in viticulture for a total of about 5,000 hectares of land. These realities make Sagrivit a true ambassador for the promotion and enhancement of excellence in the area (see also the web site which is also present among the references: https://www.sagrivit.it/)
The following sentence has been added:
…. as well as four wine estates specialized in viticulture for a total of about 5,000 hectares of land.
Line 107: As you included a reactive section, you can avoid the information in brackets in the following.
Thank you. We tried to write the information in brackets only where necessary.
Line 112: There is a strange symbol in the word ‘Scientic’ in my version.
Corrected
Lines 110-115: These methods have to be described with more detail or references have to be added.
The reference has been added.
Lines 133-134: Please, be consistent in the way you express the different steps in the gradient elution.
Experimental condition about gradient elution is given in the cited reference. According to the editor's recommendation, this part of the text has been shortened.
Line 183. There is a missing full-stop.
Corrected
Results and discussion
This section is, in general well, addressed. However, I strongly suggest the use of ANOVAs for looking for significant differences between the different groups of samples and analysis of contrasts for identifying which groups are or are not different.
At this purpose we have already answered before.
Line 205: This sentence is a bit confusing. We have partly modified the text
Lines 207-208: Have you developed an Analysis of Variance (ANOVA). If not, it is not adequate to use this expression. Moreover, it would be really interesting to develop different ANOVAs to check if there are significant differences between regions or between different wines in a same region. Or even between your results and results of other authors.
We have slightly modified the text hoping that in this way our objective is clearer. We just made some comparison on the base of the data available in the literature without applying any statistical test but simply discussing the results.
Line 211: For example, is it this difference statistically significant? Adding ANOVAs and analysis of contrasts, such as Tuckey test, will increase a lot its scientific value of the study.
The term significantly has been deleted
We agree with the reviewer that the application of this kind of test would be useful; we are grateful for this suggestion and surely will take it into consideration for future studies. Nevertheless, in this section our aim was just to compare what we have found in the examined Italian wines with what found in wines from other countries. If in the future we will have a more significant data set for Italian wines we will try this comparison but this will be another work.
Line 218: Please, moderate this type of expressions. Although the number of wines taken in to account in your study is adequate, I think it is not enough to say ‘Italian wines’. Better say ‘The Italian wines taken in to account in this study’ or ‘The studied Italian wines’. It has been corrected
Table 2: You did each measurement in triplicate so, please add the standard errors of the means reported. Perhaps in a supplementary table.
Tables with standard deviation are inserted in the supplementary material (see Table S1 and S3)
Figure 2 caption: Please describe the data shown in figures 2a and 2b.
The figure caption has been completed
Lines 264-268: If you want to highlight these relationships between samples of the same variety in a same region you can measure the distance between sample in the PCA. You can use Euclidean or Mahalanobis distance.
We agree with the referee and HCA (hierarchical cluster analysis) has been also tried; it was performed using Euclidean distance; the obtained dendrogram was not reported since the separation among the groups was not so satisfactory such as what obtained by PCA and reported in figure 2.
Lines 272-279: For me, an ANOVA would be really useful for explain this. Even if there are no significant differences, it would be interesting to know.
We have applied ANOVA one-way but the results were not significant. On the contrary it was significant was obtained applying F test to couples. In the following you can find the results where in all cases F was >F crit indicating a significant difference between these types of wines. This results have been also mentioned in the text.
test F |
MERLOT/PINOT |
|||
Test F a due campioni per varianze |
||||
|
MERLOT |
PINOT |
||
Media |
120,7585 |
208,96325 |
||
Varianza |
76362,46777 |
188418,437 |
||
Osservazioni |
20 |
20 |
||
gdl |
19 |
19 |
||
F |
0,405281294 |
ok |
||
P(F<=f) una coda |
0,027967206 |
|||
F crtitico una coda |
0,378032625 |
|
||
|
||||
test F |
PINOT/RIBOLLA |
|
||
|
||||
|
PINOT |
RIBOLLA |
|
|
Media |
208,96325 |
168,32125 |
|
|
Varianza |
188418,4365 |
101481,184 |
|
|
Osservazioni |
20 |
20 |
|
|
gdl |
19 |
19 |
|
|
F |
1,856683464 |
|
||
P(F<=f) una coda |
0,093292411 |
ok |
|
|
F crtitico una coda |
1,822402762 |
|
|
|
The same approach has been also tried for red wines obtaining the following results; it is interesting to notice that W7 and W8 were significantly difference while this difference was not present if samples W8 and W13 are compared, being both Umbria wines and for this reason very similar.
test F |
W7/W8 |
||
|
W7 |
W8 |
|
Media |
120,7585 |
229,3045 |
|
Varianza |
76362,46777 |
207393,037 |
|
Osservazioni |
20 |
20 |
|
gdl |
19 |
19 |
|
F |
0,368201695 |
||
P(F<=f) una coda |
0,017545288 |
ok |
|
F crtitico una coda |
0,330321039 |
|
|
test F |
W8/W13 |
||
Test F a due campioni per varianze |
|||
|
W8 |
W13 |
|
Media |
229,3045 |
198,457 |
|
Varianza |
207393,037 |
159417,281 |
|
Osservazioni |
20 |
20 |
|
gdl |
19 |
19 |
|
F |
1,300944514 |
the difference is not significant |
|
P(F<=f) una coda |
0,286003264 |
(in fact come both from Umbria) |
|
F crtitico una coda |
1,999222735 |
|
|
Line 352: I think there is a problem with the cites here and in the remaining manuscript. ¿Why is the author name added to each cite? corrected
Table 3: Again, add the standard errors.
Table with standard errors are inserted in the supplementary material (table S1 and S3).
Round 2
Reviewer 1 Report
The paper has been improved as I requested.
Author Response
We thank the reviewer for his comments since in this way the paper has been improved.
Reviewer 2 Report
The paper was improved according to my suggestions however there are still some minor corrections that need to be done before publication.
Line 55 - I don't really understand why do you want to postulate wine as a funcional food. Polyphenolic compounds associated with the health benefits present in wines are also present in other non-alcoholic food matrices and the alcohol in wine is bad for health. I know that a moderate wine consumption has some benefits but I think the term functional food it is not the most accurate to define an alcoholic beverage. Moreover, you say that wines could be postulated as functional foods but for instance white wines are very poor in polyphenolic compounds and by this you are also generalizing which is not correct.
Line 149: You should include "...analysis of anthocyanins and anthocyanin-derived pigments"
You say that through LC / HRMS technique, you cannot know exactly whether they are glucose or galactose derivatives. But if anthocyanins present in wines are glucosides, the the derivatives that are formed during aging are also glucosides. If you used standards for anthocyanins you know that you have those anthocyanins and by this the derivatives present are also glucosides.
Do you include references from the authors that identified those pigments that you refer in Table 4 or Table S2. For instance, peak 13 was identified for the first time by "Fulcrand, H.; dos Santos, P.-J. C.; Sarni-Manchado, ; Cheynier, V.; Favre-Bonvin, J. Journal of the Chemical Society, Perkin Transactions 1 1996, 735-739."
Author Response
We thank the reviewer for his further comments. We have answered to the further questions (in attach also the pdf version of the file) and we hope that now it is more clear.
Reviewer 2
The paper was improved according to my suggestions however there are still some minor corrections that need to be done before publication.
Line 55 - I don't really understand why do you want to postulate wine as a funcional food. Polyphenolic compounds associated with the health benefits present in wines are also present in other non-alcoholic food matrices and the alcohol in wine is bad for health. I know that a moderate wine consumption has some benefits but I think the term functional food it is not the most accurate to define an alcoholic beverage. Moreover, you say that wines could be postulated as functional foods but for instance white wines are very poor in polyphenolic compounds and by this you are also generalizing which is not correct.
We thank the reviewer for this comment and we agree with him on the fact that red wines, having a higher polyphenolic content, bring to some more benefit for human health with respect to white wines.
We have modified the sentence in this way eliminating the term funcional:
Synergistic effects of individual polyphenols present especially in red wine, could result in positive impact on human wellbeing [7].
In fact, the definition functional food was reported in ref 7 but in any case referring at these two other papers:
Kallithraka, S., Tsoutsouras, E., Tzourou, E., Lanaridis, P., 2006. Principal phenolic compounds in Greek red wines. Food Chem. 99, 784–793.
a Torre, G.L., Saitta, M., Vilasi, F., et al., 2006. Direct determination of phenolic compounds in Sicilian wines by liquid chromatography with PDA and MS detection. Food Chem. 94, 640–650.
Line 149: You should include "...analysis of anthocyanins and anthocyanin-derived pigments"
We agree with the reviewer, it has been corrected
You say that through LC/HRMS technique, you cannot know exactly whether they are glucose or galactose derivatives. But if anthocyanins present in wines are glucosides, the derivatives that are formed during aging are also glucosides. If you used standards for anthocyanins you know that you have those anthocyanins and by this the derivatives present are also glucosides.
We agree with this comment, the “hexoside” has been changed to “glucoside” throughout the whole text, as well as in the supplementary Table S4.
Do you include references from the authors that identified those pigments that you refer in Table 4 or Table S2. For instance, peak 13 was identified for the first time by "Fulcrand, H.; dos Santos, P.-J. C.; Sarni-Manchado, ; Cheynier, V.; Favre-Bonvin, J. Journal of the Chemical Society, Perkin Transactions 1 1996, 735-739."
We have added at the end of the text that some of the substances reported, for instance peak 13, have been previously identified (ref 49).

Reviewer 3 Report
The authors have addressed almost all my suggestions. They have also correctly explained why the remaining suggestions are not addressed.
They have tried to apply ANOVA and F-test to the data. Although it was not really useful in all cases, I think the manuscript has gained scientific soundness.
Author Response
We thank very much the reviewer for the comments. We also think that now the paper, adding these further data treatments, has been improved gaining more scientific soundness.